# ReAttention: Training-Free Infinite Context with Finite Attention Scope

**Xiaoran Liu**[1,3,4,*] **Ruixiao Li**[1,4,*] **Zhigeng Liu**[1,*] **Qipeng Guo**[3,4,*] **Yuerong Song**[1,4],
**Kai Lv**[1,3], **Hang Yan**[3], **Linlin Li**[2], **Qun Liu**[2], **Xipeng Qiu**[1,3,4]
[1]School of Computer Science, Fudan University, [2]Huawei Noah's Ark Lab,
[3]Shanghai AI Lab, [4]Shanghai Innovation Institute
`xrliu24@m.fudan.edu.cn, xpqiu@fudan.edu.cn`

## Abstract

The long-context capability of the Large Language Models (LLM) has made significant breakthroughs, but *the maximum supported context length in length extrapolation* remains a critical bottleneck limiting their practical applications. The constraint of context length in LLMs arises from the self-attention mechanism, which cannot effectively and efficiently capture the semantic relationships within infinitely long contexts via the limited pre-trained positional information and attention scope. In this work, we propose **ReAttention**, a training-free approach enabling LLM based on the self-attention mechanism to support an infinite context with a finite attention scope under sufficient memory resources. ReAttention performs the position-agnostic top-$k$ attention before the ordinary position-aware self-attention, freeing LLMs from the length extrapolation issue. We validate the performance of ReAttention on the LongBench, L-Eval, and InfiniteBench and demonstrate that it is on par with traditional methods. Furthermore, we also apply ReAttention on mainstream LLMs, including LLaMA3.1-8B and Mistral-v0.3-7B, enabling them to support context lengths of at least 1M and even expanding the context length of LLaMA3.2-3B-chat by $128\times$ to 4M without any further training in Needle-In-A-Haystack tests. We also improve the efficiency of ReAttention with Triton and achieve an efficient extrapolation without additional overhead. The code is available at `https://github.com/OpenMOSS/ReAttention`.

## 1 Introduction

Large Language Models (LLM) based on Transformer (Vaswani et al., 2017; OpenAI, 2023; Reid et al., 2024; Sun et al., 2024) has made great progress in the applications of Natural Language Processing (NLP). Particularly, in long context modeling, a considerable amount of research has been dedicated to extending the length capabilities of LLMs (Chen et al., 2023; Peng et al., 2023; Liu et al., 2023; Xiong et al., 2023), increasing the maximum supported context length from the initial 2K (Touvron et al., 2023) to 2M (Ding et al., 2024) in open-source LLMs. However, *the maximum context length in length extrapolation* remains a bottleneck that limits the practical applications of LLMs (Press et al., 2022; Chen et al., 2023). To achieve infinite context length under sufficient memory for Transformer-based LLMs, the following three conditions must be satisfied:

- a *The position information in the inference phase should not be out-of-distribution (OOD) compared to the training phase;* (Han et al., 2023; Liu et al., 2023)
- b *The self-attention entropy in the inference phase should not increase with the length of the input;* (Han et al., 2023; Xiao et al., 2023)
- c *LLM should keep effectively aware of critical contextual information at each inference step.* (Dong et al., 2024; Zhang et al., 2024a)

Early works in extrapolation focus on the first point, primarily by interpolating the position embedding (Chen et al., 2023; bloc97, 2023b;a; Xiong et al., 2023) or limiting the relative position

---

*\* Equal contribution. This work is done during Xiaoran Liu's internship in Shanghai AI Lab*

within the pre-training context length (Su, 2023; Jin et al., 2024; An et al., 2024). However, later studies have shown that even if positional information is not OOD, the attention entropy tends to increase logarithmically with the length of the self-attention window (Han et al., 2023). Specifically, the self-attention distribution becomes increasingly diffuse as inference length expands, making it difficult to effectively gather information from the context, which leads to unstable model outputs (Peng et al., 2023; Han et al., 2023). In response, methods like LM-Infinite (Han et al., 2023) and StreamingLLM (Xiao et al., 2023) have been proposed, which retain the starting and ending parts of the input, allowing LLMs to maintain stable outputs as input length increases. However, they compromise the global awareness of contextual information, harming the downstream performance.

For human beings, although we have long-term memory, we only need a small amount of information to think and act in real-time. Similarly, while an LLM requires the full context to complete the inference process, it only needs limited contextual information at each inference step. This fact has inspired research like InfLLM (Xiao et al., 2024a) and LongHeads (Lu et al., 2024) to perceive and extract critical information from the context before performing self-attention. However, LongHeads still face upper limits on positional information due to the use of NTK or PI for prefill (Lu et al., 2024), while InfLLM suffers from bias in representing contextual information, raised from the chunk-wise representation and ReRoPE-style position embedding when extracting key information.

In this work, we introduce **ReAttention**, a training-free method that extends LLMs with finite context lengths to process infinite contexts. ReAttention regards extracting critical contextual information as an additional attention process preceding traditional self-attention, akin to "thinking before you act." At each reasoning step, ReAttention selects the most critical finite segments from the KV cache without position information, concatenates them, applies positional embedding, and performs self-attention. By controlling the length of the selected KV cache segments, ReAttention can achieve infinite context length with a finite attention scope, while ensuring the position information and attention entropy are not out-of-distribution (Han et al., 2023). Furthermore, inspired by the optimization techniques used in self-attention, particularly FlashAttention (Dao et al., 2022; Dao, 2023), we use Triton (Tillet et al., 2019), a GPU programming language, to minimize read and write overheads in top-$k$ attention. With our custom Triton kernel, ReAttention avoids the extra computational overhead and reduces memory usage for long contexts. Our contributions are summarized as follows:

- We outline three requirements for infinite context in Transformer-based LLMs, position embedding not OOD, stable attention entropy, and effective contextual awareness. We also find that the last one can be satisfied via the attention score without positional embedding.
- Based on this observation, we propose a training-free method, ReAttention, satisfying the aforementioned three conditions, thus extrapolating the context length of LLMs to infinity with a finite attention scope, and making LLMs free from the issue of length extrapolation.
- We validate that ReAttention matches the performance of traditional self-attention in long contexts with no computational overhead and less memory usage. Specifically, ReAttention extends the context length of leading LLMs, such as LLaMA3.1-8B-128K, to at least 1 million tokens. For smaller models like LLaMA3.2-3B-chat, the context length can be increased by $128\times$, reaching up to 4 million tokens, without additional training.

## 2 METHOD

The overall structure of ReAttention is illustrated in Figure 1, which consists of a position-agnostic top-$k$ attention responsible for full-context cache selection and a traditional self-attention transform with position embedding. ReAttention achieves a training-free integration between the two components.

### 2.1 FULL-CONTEXT CACHE SELECTION

While LLMs require a complete long context to perform the entire inference process, only a limited context segment is needed at each inference step (Xiao et al., 2023; 2024a; Lu et al., 2024). Considering the beginning and end of the input context correspond to globally important prompts and local information for inference (Xiao et al., 2023), ReAttention retains both the global and local segments of the KV cache for inclusion in the self-attention process.

$$\mathcal{K}_{\text{cache}} = [\mathcal{K}_{\text{global}}, \mathcal{K}_{\text{middle}}, \mathcal{K}_{\text{local}}], \qquad \mathcal{V}_{\text{cache}} = [\mathcal{V}_{\text{global}}, \mathcal{V}_{\text{middle}}, \mathcal{V}_{\text{local}}]. \qquad (1)$$

Then, ReAttention uses the query vector of the current step to perform a top-$k$ selection on the middle part of the KV cache (Ribar et al., 2023), to identify the most important cache segments for the current step as shown in Figure 1.

$$\text{Indices} = \text{top-}k\left(\boldsymbol{q}_t \mathcal{K}_{\text{middle}}^T\right),$$
$$\mathcal{K}_{\text{select}} = \mathcal{K}_{\text{middle}}[\text{Indices}], \tag{2}$$
$$\mathcal{V}_{\text{select}} = \mathcal{V}_{\text{middle}}[\text{Indices}].$$

ReAttention performs a full-context selection on the KV cache in each layer, allowing different layers to choose different KV caches for the calculation. Furthermore, since each attention layer has multiple attention heads and ReAttention adopts chunked streaming input during the prefilling stage, multiple query vectors may exist simultaneously. In this case, ReAttention

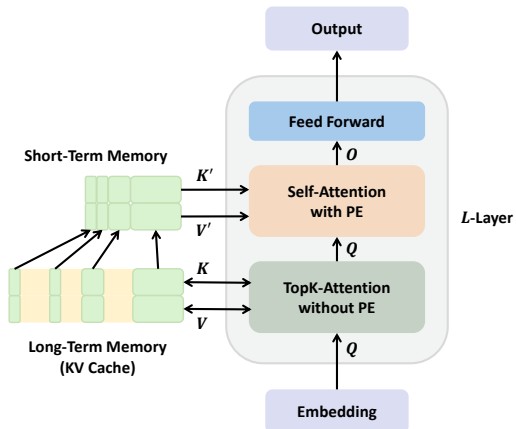

Figure 1: Overview of ReAttention.

votes based on the top-$k$ selections from different heads and query vectors to identify the top-$k'$ KV caches. Additionally, to ensure semantic coherence, ReAttention not only retains the top-$k'$ elements themselves but also extracts $m$ neighboring entries as a whole. Overlapping parts are deduplicated.

Unlike chunk-based selection in previous works (Lu et al., 2024; Xiao et al., 2024a), ReAttention extracts critical contextual information with the dot-product between $\boldsymbol{q}_t$ and $\mathcal{K}_{\text{cache}}$, rather than a chunk-wise representation of $\mathcal{K}_{\text{cache}}$. This method offers strong adaptability and avoids the semantic fragmentation caused by fixed chunking (Luo et al., 2024). Moreover, since the intermediate result of $\boldsymbol{q}_t \mathcal{K}_{\text{middle}}^T$ is too large for long-context scenarios, and would incur huge read and write overhead, we follow the approach of FlashAttention (Dao et al., 2022; Dao, 2023), fusing the whole process into one kernel using Triton (Tillet et al., 2019), as detailed in Section 3.4.

## 2.2 TRAINING-FREE INTEGRATION

ReAttention concatenates the selected KV cache segments between the global and local parts, applies positional embedding sequentially, and preserves the relative order while ignoring the absolute distance between the selected segments, as shown in Figure 1. Self-attention can then be applied to the concatenated KV cache. The pseudocode of the whole process is detailed in Appendix A.

$$\mathcal{K}_{\text{cache'}} = [\mathcal{K}_{\text{global}}, \mathcal{K}_{\text{select}}, \mathcal{K}_{\text{local}}], \qquad \mathcal{V}_{\text{cache'}} = [\mathcal{V}_{\text{global}}, \mathcal{V}_{\text{select}}, V_{\text{local}}],$$
$$\tilde{\boldsymbol{q}}_t, \tilde{\mathcal{K}}_{\text{cache'}} = \text{PE}\left(\boldsymbol{q}_t, \mathcal{K}_{\text{cache'}}\right), \qquad \boldsymbol{o}_t = \text{SelfAttn}\left(\tilde{\boldsymbol{q}}_t, \tilde{\mathcal{K}}_{\text{cache'}}, \mathcal{V}_{\text{cache'}}\right). \tag{3}$$

It's important to note that, unlike the conventional implementation in Huggingface Transformers (Wolf et al., 2020), where the position embedding is applied before KV caching, in ReAttention, the position embedding is separated from the KV cache and performed after the KV cache selection. That is to say, the cached KV does not include the positional information. This design offers several advantages. On the one hand, as mentioned in Section 4, the attention score without positional embedding is more conducive to locating the key information in the context. On the other hand, since the concatenated cache length remains within the pre-training context length or the extrapolation upper bound, the position embedding will never be OOD (Han et al., 2023).

Furthermore, the unselected KV cache segments are unimportant in the current inference step, as their self-attention scores are minimal (Zhang et al., 2024d). Therefore, this modification does not harm the self-attention output and can eliminates interference from irrelevant information (Zhang et al., 2024d; Ge et al., 2023). Compared with InfLLM (Xiao et al., 2024a), ReAttention's position embedding and attention transform within the pre-training window do not introduce the untrained attention pattern, thereby preventing bias accumulation in the KV cache during the prefilling stage. Thus, ReAttention can disregard position information, achieve an unlimited attention context with a limited attention scope without any training, and remain compatible with the existing attention acceleration methods (Dao et al., 2022; Dao, 2023).

|  | S-Doc | M-Doc | Sum | ICL | Syn. | Code | Avg. |
|---|---|---|---|---|---|---|---|
| *LLaMA3-8B-8K* | 18.08 | 13.17 | 21.42 | 63.95 | 10.45 | 69.89 | **35.48** |
| + StreamingLLM | 15.14 | 8.90 | 18.36 | 58.19 | 7.28 | 70.27 | 32.49 |
| + InfLLM | 12.43 | 8.93 | 13.20 | 66.08 | 7.55 | 70.03 | 32.51 |
| + ReAttention (Ours) | 19.08 | 11.33 | 20.69 | 63.40 | 8.95 | 70.38 | 35.03 |
| *LLaMA3.1-8B-128K* | 23.61 | 15.29 | 16.15 | 64.61 | 31.81 | 70.63 | **38.79** |
| + StreamingLLM | 21.67 | 11.46 | 14.01 | 61.05 | 33.43 | 70.50 | 37.06 |
| + InfLLM | 17.11 | 9.70 | 15.12 | 68.46 | 34.30 | 68.37 | 37.14 |
| + ReAttention (Ours) | 23.97 | 15.22 | 15.66 | 64.54 | 31.20 | 70.48 | 38.63 |
| *LLaMA3.2-3B-128K* | 34.45 | 26.24 | 24.46 | 59.29 | 40.11 | 55.73 | **40.76** |
| + StreamingLLM | 28.05 | 19.41 | 22.67 | 55.23 | 22.63 | 54.16 | 35.08 |
| + InfLLM | 17.47 | 6.77 | 17.44 | 49.11 | 23.63 | 48.09 | 28.21 |
| + ReAttention (Ours) | 32.53 | 24.39 | 24.13 | 58.13 | 37.39 | 55.17 | 39.43 |

Table 1: Results of LLaMA Series(Meta, 2024a; Dubey et al., 2024; Meta, 2024b) on LongBench(Bai et al., 2023). ReAttention achieves a consistent superiority over StreamingLLM(Xiao et al., 2023) and InfLLM(Xiao et al., 2024a) and shows comparable performance with LLMs with full attention.

|  | 32K | | | 64K | | | 128K | | | Avg. |
|---|---|---|---|---|---|---|---|---|---|---|
|  | MC | QA | Sum | MC | QA | Sum | MC | QA | Sum | |
| *LLaMA3-8B-Instruct-8K* | 50.66 | 7.44 | 21.76 | - | - | - | - | - | - | - |
| + StreamingLLM | 27.95 | 4.40 | 18.41 | 28.38 | 4.30 | 17.61 | 34.93 | 4.36 | 17.64 | 17.55 |
| + InfLLM | 40.61 | 4.15 | 20.30 | 35.37 | 5.12 | 18.26 | 33.19 | 5.78 | 19.70 | 20.28 |
| + ReAttention (Ours) | 46.29 | 5.33 | 21.76 | 44.54 | 5.25 | 20.15 | 49.78 | 5.98 | 20.36 | **24.38** |
| *LLaMA3.1-8B-Instruct-128K* | 37.12 | 18.85 | 1.60 | 31.88 | 23.32 | 1.39 | 21.83 | 25.43 | 1.40 | 18.09 |
| + StreamingLLM | 34.50 | 6.82 | 19.61 | 34.93 | 6.83 | 20.82 | 35.81 | 6.71 | 20.10 | 20.68 |
| + InfLLM | 32.31 | 5.48 | 21.00 | 34.50 | 7.56 | 19.82 | 39.30 | 8.92 | 19.27 | 20.91 |
| + ReAttention (Ours) | 36.68 | 12.71 | 19.68 | 35.37 | 12.14 | 18.84 | 40.61 | 12.63 | 18.14 | **22.98** |
| *LLaMA3.2-3B-Instruct-128K* | 14.41 | 15.65 | 1.38 | 11.79 | 18.16 | 1.27 | 13.10 | 17.63 | 1.30 | 10.52 |
| + StreamingLLM | 19.21 | 8.41 | 18.81 | 19.65 | 8.91 | 18.73 | 20.09 | 8.74 | 18.80 | **15.71** |
| + InfLLM | 20.09 | 4.35 | 13.66 | 23.14 | 9.38 | 12.52 | 22.27 | 12.08 | 14.25 | 14.64 |
| + ReAttention (Ours) | 19.21 | 11.80 | 16.75 | 17.47 | 11.82 | 15.06 | 20.09 | 11.91 | 14.13 | 15.36 |

Table 2: Results of LLaMA Series on InfiniteBench (Zhang et al., 2024b) in different context lengths. "-" means LLM could not provide a stable output in a certain context length. ReAttention achieves superiority over StreamingLLM (Xiao et al., 2023), InfLLM (Xiao et al., 2024a) and full attention.

# 3 EXPERIMENT

## 3.1 SETUP

We conduct experiments on LLaMA3-8B-8K (Meta, 2024a), LLaMA3.1-8B-128K (Dubey et al., 2024), LLaMA3.1-70B-128K (Dubey et al., 2024), LLaMA3.2-3B-128K (Dubey et al., 2024), Mistral-v0.3-7B-32K (mistralai, 2024), InternLM2.5-7B-1M (InternLM, 2024), Qwen2-7B-128K (Yang et al., 2024a), Qwen2-72B-128K (Yang et al., 2024a), Qwen2-1B-32K (Yang et al., 2024a). For all models, we set the length of $\mathcal{K}_{\text{global}}$ to 32, the length of $\mathcal{K}_{\text{local}}$ to 4096, and selected span size to 32. Moreover, we set $k = 4, k' = 127$ in top-$k$ attention. Importantly, the attention scope in each step remains within the maximum attention window. For example, for the LLaMA3-8B-8K with ReAttention, the maximum attention scope size is $32 + 4096 + 127 \times 32$, which exactly matches the maximum supported attention window of 8192. We use OpenCompass (Contributors, 2023b) for validation. All experiments are performed with FP16 precision and accelerated with FlashAttention2 (Dao, 2023).

## 3.2 LONG-CONTEXT BENCHMARK EVALUATION

We first evaluate all 9 LLMs on the commonly used long-context benchmark LongBench (Bai et al., 2023) and L-Eval (An et al., 2023), with a default context length of 32K and a middle truncation.

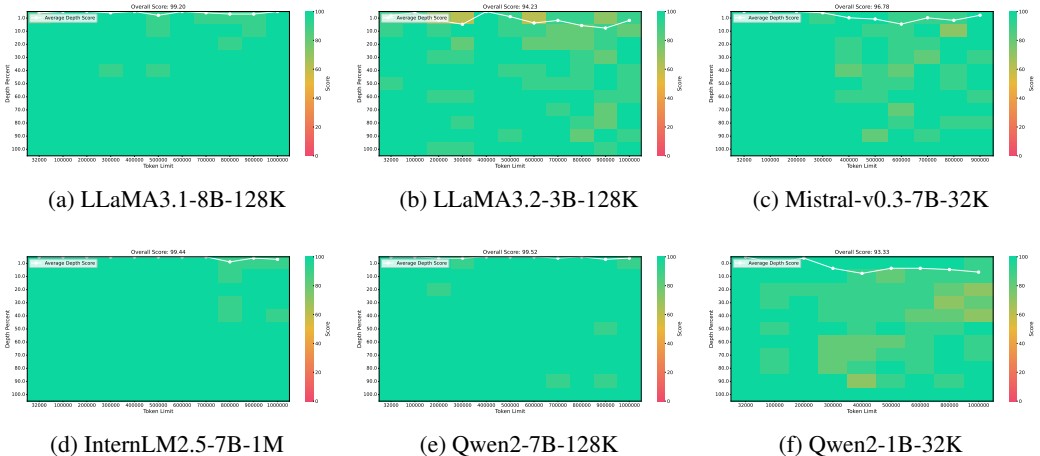

(a) LLaMA3.1-8B-128K     (b) LLaMA3.2-3B-128K     (c) Mistral-v0.3-7B-32K

(d) InternLM2.5-7B-1M     (e) Qwen2-7B-128K     (f) Qwen2-1B-32K

Figure 2: Results of ReAttention-enhanced existing mainstream LLMs, including LLaMA3-8B-8K and Mistral-v0.3-7B-32K, on Needle-In-A-Haystack (Contributors, 2023a) implemented in OpenCompass (Contributors, 2023b).

For LLaMA3-8B-8K, which has a context length of less than 32K, we report its performance with Dynamic NTK (bloc97, 2023a). For the Dynamic NTK implementation, we use the default settings from the Huggingface Transformers (Wolf et al., 2020), setting the scaling factor to 4. Additionally, we compare the performance of all 9 LLMs with StreamingLLM (Xiao et al., 2023) as an ablation study, using the same global and local segment lengths as the ReAttention settings.

As shown in Table 1 and supplemented in Table 5 in Appendix B, ReAttention outperforms StreamingLLM across all 9 models, indicating that the full-context selection acquires useful information for long-context inference. Furthermore, ReAttention performs on par with full attention and even surpasses it in some cases, such as LLaMA3.1-70B-128K (Meta, 2024a) and Qwen2-1B-32K (Yang et al., 2024a). This demonstrates that ReAttention can be applied to LLMs of various sizes and achieve performance close to full attention on downstream tasks (Jiang et al., 2024).

To further demonstrate the superiority and extrapolation capability of ReAttention, we validate our method on InfiniteBench (Zhang et al., 2024c), a more challenging benchmark with a longer context length. We choose 3 commonly tested subtasks, En.MC, En.QA and En.Sum, evaluate models with varying context lengths, and compare ReAttention with DynamicNTK (bloc97, 2023a) and InfLLM (Xiao et al., 2024a) with the same selection config. The results[1] are shown in Table 2. Remarkably, ReAttention consistently outperforms full-attention and InfLLM at the 128K context length and in the average score. While DynamicNTK performs well at 32k, it encounters a clear upper bound on extrapolation, beyond which the model fails to produce stable outputs. Moreover, while InfLLM can extend context length indefinitely (Xiao et al., 2024a), it still lags behind ReAttention in downstream tasks due to inaccurate extraction of critical information and differences in position embedding format compared to the pre-training phase.

### 3.3 NEEDLE-IN-A-HAYSTACK EVALUATION

Building upon the long-context benchmark evaluation, we use models with strong retrieval capabilities within their training context lengths and conduct the Needle-In-A-Haystack (NIAH) evaluation (Contributors, 2023a;b). We perform experiments on 8 A100 GPUs and extend the context lengths of LLMs with ReAttention to at least 1M tokens. As shown in Figure 2, LLMs with ReAttention maintain remarkably high retrieval accuracy across the entire range of context lengths they could support, regardless of their original attention windows. Importantly, we also extend the context of mainstream LLMs like LLaMA3-8B-8K (Meta, 2024a) and Mistral-v0.3-7B-32K (mistralai, 2024) to at least 1M, providing the community with an effective solution for deploying long-context LLMs.

---

[1]Due to the extensive evaluation for all 9 models, and considering that InfLLM has limited support for larger models such as LLaMA3.1-70B-chat and other series like InternLM2.5 and Qwen2, we only report results of the LLaMA series models at the 8B and 3B scales.

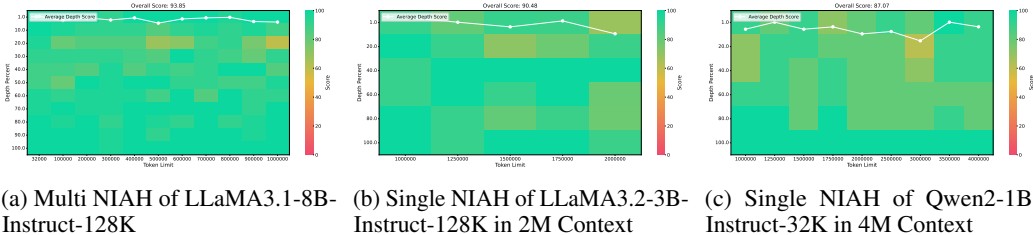

(a) Multi NIAH of LLaMA3.1-8B-Instruct-128K

(b) Single NIAH of LLaMA3.2-3B-Instruct-128K in 2M Context

(c) Single NIAH of Qwen2-1B-Instruct-32K in 4M Context

Figure 3: Results of Multi-Needle-In-A-Haystack (Reid et al., 2024) and Single NIAH in a longer context length implemented in OpenCompass (Contributors, 2023b).

Building on this, we increase the difficulty of the evaluation by conducting experiments on multi-NIAH and single-NIAH in longer contexts, with results shown in Figure 3. Given the increased difficulty, we employ the corresponding instruct versions of the models. On the mainstream LLaMA3.1-8B-Instruct-128K (Dubey et al., 2024), we achieve a context length of 1M in the multi-NIAH task. For smaller models, such as LLaMA3.2-3B-chat-128K (Meta, 2024b), we extend the context length to at least 2M. Notably, for Qwen2-1B-Instruct-32k (Yang et al., 2024a), we extend the context length to 4M on 4 A100 GPUs, achieving a training-free context expansion of $128\times$. To the best of our knowledge, it is the largest amplification of context length for LLMs achieved without additional training. This demonstrates that ReAttention can extend the context length while disregarding the position information, using a finite attention window to achieve an infinite attention context.

## 3.4 EFFICIENCY ANALYSIS

In the PyTorch framework, operators execute independently, requiring frequent I/O to the GPU memory, which introduces unnecessary overhead and latency. Inspired by FlashAttention (Dao et al., 2022; Dao, 2023), we develop a GPU kernel function for the top-$k$ attention described in Section 2.1 using Triton. As shown in Figure 4, our kernel fuses operators for attention score computation and top-$k$ calculation, enabling the entire process to run within the GPU cache. Thus GPU memory I/O is significantly reduced, improving both GPU memory usage and runtime. We added top-$k$ attention that reduces the time for the self-attention, and our kernel keeps top-$k$ attention's overhead minimal, leaving overall latency unchanged.

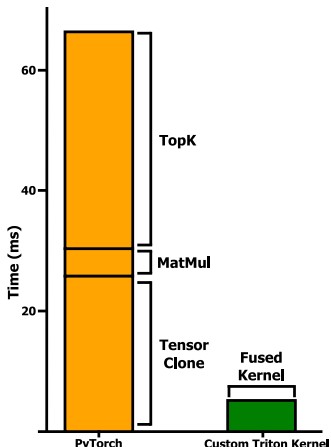

Figure 4: Overview of the kernel fusion in our customized top-$k$ attention kernel. The performance measurements reflect the execution time of the corresponding kernel functions, with the input length 8K for Llama3.1-8B inference tasks.

We analyze our kernel's efficiency in terms of GPU execution time and memory usage, showing that using Triton rather than PyTorch for the top-$k$ attention greatly enhances GPU performance while reducing GPU memory consumption. Additionally, we compare ReAttention with the standard implementation in the HuggingFace Transformers (Dai et al., 2019), measuring time to first token (TTFT) and memory cost in context lengths 32k to 256k. The TTFT and throughput comparison results are detailed in Appendix C. All experiments were conducted on a system with a 48-core CPU, 256GB RAM, and an A800-80GB GPU.

**Triton Operator Efficiency** We tested our Triton operator using real inference inputs (e.g., $\mathcal{Q}$ and $\mathcal{K}$ matrices extracted from real-world inference tasks) to measure execution time and device memory usage. As shown in Figure 5a, the standard PyTorch implementation exceeds 80GB of memory for sequences over 64k, while our Triton operator minimizes memory usage, limited to input and output matrices. It also achieves hundreds of times faster performance at 64k sequence lengths and scales efficiently as length increases (Figure 5b).

**Memory Usage in Prefilling Stage** Memory overhead limits the prefilling for longer sequences. Figure 6 shows that our method surpasses the standard implementation in HuggingFace Transformers

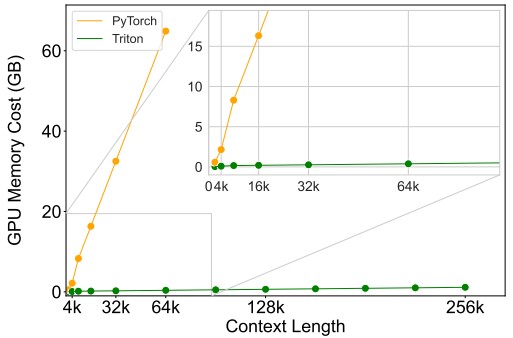
(a) GPU memory cost across different context lengths.

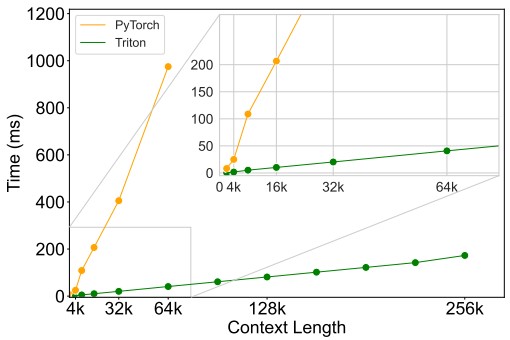
(b) Latency across different context lengths.

Figure 5: The efficiency cooperation between our Triton kernel and its PyTorch version. Our fused top-$k$ attention kernel performs better than PyTorch implementation across all context lengths.

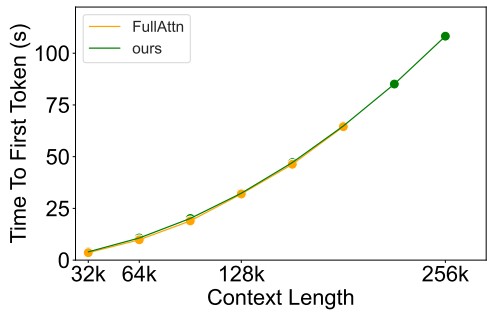

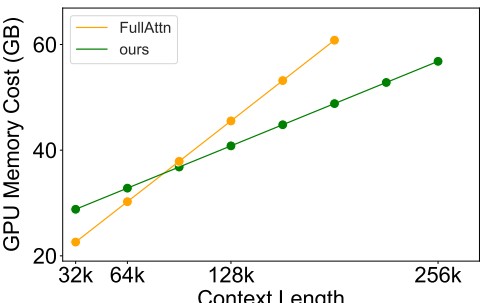

Figure 6: First token latency(FTL) and GPU memory consumption in the prefilling stage. FullAttn refers to the official HuggingFace Transformers implementation of Llama3.1-8B-Base, which runs out of memory after 192k.

as context length grows, continuing to work when the standard implementation runs out of memory. We also record first-token latency, which remains comparable to the standard implementation in Huggingface Transformers.

## 4 DISCUSSION

### 4.1 ANALYSIS ON HYPER-PARAMETER

We first discuss the selection of hyper-parameters. We evaluate the performance of the LLaMA3-8B-8K (Meta, 2024a) on the LongBench benchmark (Bai et al., 2023), comparing hyper-parameters including chunk size, span size, local size, and top-$k$ values. Since the prefilling chunk is always contained within the local part in self-attention, the chunk size must be smaller than the local size. Additionally, to ensure fairness in our comparisons, we

|  | En. | Zh. | Code | Avg. |
|---|---|---|---|---|
| StreamingLLM | 29.48 | 13.22 | 70.27 | 32.49 |
| ReAttention | 33.42 | 18.74 | 70.38 | **35.03** |
| + chunk size = 1024 | 25.50 | 18.07 | 70.32 | 34.75 |
| + chunk size = 2048 | 25.39 | 18.07 | 70.30 | 34.72 |
| + span size = 8 | 13.02 | 8.74 | 58.54 | 22.96 |
| + span size = 16 | 14.13 | 9.43 | 60.92 | 24.29 |
| + span size = 64 | 25.27 | 17.98 | 70.83 | 34.69 |
| + span size = 128 | 25.23 | 16.97 | 70.54 | 34.57 |
| + top-$k$ = 1 | 25.04 | 16.57 | 70.75 | 34.45 |
| + top-$k$ = 8 | 25.40 | 17.51 | 70.64 | 34.70 |
| + local size = 2048 | 24.51 | 16.77 | 70.39 | 34.08 |
| + local size = 1024 | 22.31 | 13.64 | 69.48 | 32.23 |

Table 3: Analysis of ReAttention hyper-parameters on LLaMA3-8B-8K and LongBench, by default chunk size 512, span size 32 as well as global size, local size 4096, and top-$k$ Selection.

maintain a consistent maximum attention window size across different settings, specifically keeping the summation of global size, span size times $k'$, and local size equal to 8192. To achieve this, we set global size equal to span size, retaining the earliest portion as the first selection segment. As

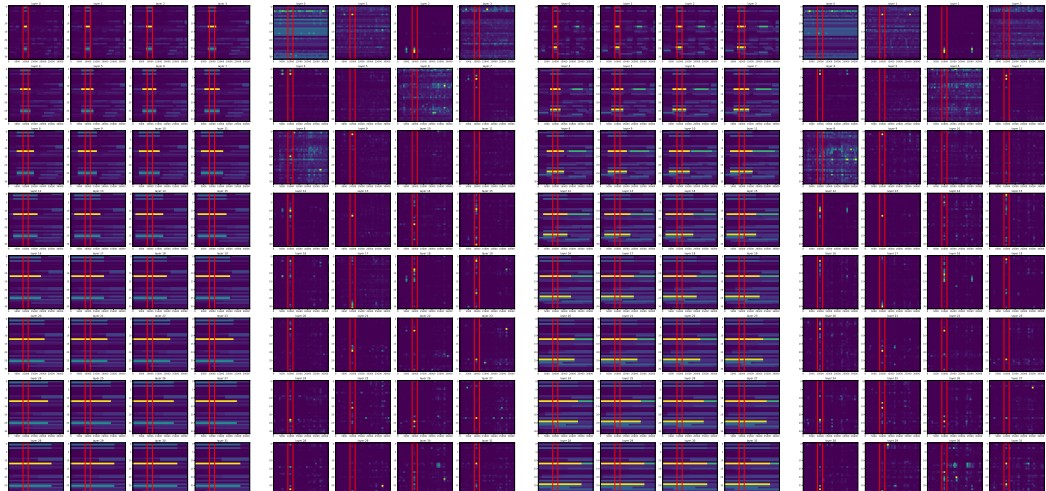

(a) Attention distribution of a correct case with position embedding.

(b) Attention distribution of a correct case without position embedding.

(c) Attention distribution of a wrong case with position embedding.

(d) Attention distribution of a wrong case without position embedding.

Figure 7: Visualization of the attention distributions for InternLM2-7B-200K (Cai et al., 2024b) evaluated on Needle-In-A-Haystack (Contributors, 2023a;b) in 32K context length. In each sub-figure, each heatmap represents the attention distribution of a single layer, with the y-axis corresponding to the 32 attention heads and the x-axis corresponding to the 32K context length. The first 32 tokens, the instruction part, and the last 6 tokens, the generated part, are not excluded. The values represent the cumulative self-attention distributions for each token generation. The brighter the color, the higher the cumulative attention. A pooling operation with a kernel size of 100 is applied sequentially to show the pattern more clearly. The red box represents the position of the "needle" in the context.

span size increased, we correspondingly reduced the $k'$ to limit the upper bound of cache selection. Additionally, when local size decreases, $k'$ is increased. The final results are shown in Table 3.

We find that as the chunk size decreases from 2048 to 512, the performance gradually improves. The prefilling stage becomes more precise in extracting critical information during each top-$k$ attention step. However, since prefilling takes more time as chunk size decreases, we select 512 as our default value without further reduction. Next, we examine the impact of different span sizes. Among these hyper-parameters, span size has the most significant effect on downstream performance. When span size is relatively small, fragmented segments weaken coherence and may mislead the LLM's predictions, resulting in poorer performance compared to StreamingLLM. When the span size reaches 32, the performance on LongBench peaks. Beyond this point, due to the constraint of pre-trained position embedding, the number of extracted segments decreases, hindering the effective capture of critical contextual information. We also compare the influence of the top-$k$ number and find that top-4 yields the best results. A smaller top-$k$ number makes it difficult for LLM to identify critical information, while a larger top-$k$ number may introduce irrelevant content that interferes with judgment. Finally, we analyze the effect of local size. Smaller local size compromises LLM's ability to maintain semantic coherence, and excessive selection beyond the local context may skew the KV cache during the prefilling stage. Therefore, we set the local size to half of the pre-training context length for LLaMA3-8B-8K and apply this hyper-parameter configuration to other LLMs.

## 4.2 POSTION-AGNOSTIC CACHE SELECTION

In ReAttention, the full-context cache selection is based on the dot product between the query and key vectors without position embedding. This may create a gap between the selection of the KV cache and the self-attention distribution. Therefore, here raises the question of whether using semantic vectors without position information can effectively locate the critical information in the context. To analyze this, we select a correct case and a wrong case from the results of Needle-In-A-Haystack evaluation (Contributors, 2023a;b) on InternLM2-7B-200K (Cai et al., 2024b) with full attention in

32K context length. For each case, we calculate the attention distribution with position embedding, i.e., the real self-attention distribution, and the attention distribution without position embedding.

The result is shown in Figure 7. For the correct case, the attention distributions, both with and without position embedding, can locate the position of the "needle" in the "haystack". However, the attention distribution with position embedding appears more dispersed, and this effect becomes more pronounced in the higher layers of the model. For the wrong case, the attention distribution with position embedding is also very dispersed and highly influenced by a particular attention head misled by irrelevant information from the lower layers. This ultimately results in the model's failure.

Interestingly, when observing the attention distribution without position embedding, we find that the model can identify the position of the "needle" using the inner product of semantic vectors and effectively filter out a large amount of irrelevant noise. Practically, the ReAttention-enhanced InternLM2-7B-200K can successfully locate the "needle" in this case. Therefore, the inner product of semantic vectors in the cache selection is reasonable, and it proves to be more effective in finding the relevant context compared with position-aware inner products. Additionally, due to the absence of position embedding in the KV cache, ReAttention can achieve an effective context extension.

## 4.3 Observations on Synthetic Tasks

The above analysis has demonstrated ReAttention's good performance in various long-context scenarios, suggesting that it can extrapolate to infinite context lengths. However, these evaluations primarily focus on common natural contexts and lack currently widely-discussed benchmarks (Kuratov et al., 2024; Li et al., 2024), such as RULER (Hsieh et al., 2024). Unfortunately, ReAttention performs poorly on the RULER benchmark. In fact, the extrapolation methods not based on full attention, including ReAttention and InfLLM, fail to pass the RULER benchmark. For instance, using the LLaMA3-8B-8K (Meta, 2024a), we compare Dynamic NTK extrapolation with InfLLM (Xiao et al., 2024a) and ReAttention at 8K and 16K context lengths. The results are shown in Table 4. Only Dynamic NTK achieves effective extrapolation, while both ReAttention and InfLLM exhibit more pronounced performance degradation as the context length increases. Specifically, We focus on reporting the results of two subtasks from RULER: NIAH-Single3 and NIAH-MultiKey3 (Hsieh et al., 2024). Both tasks involve extracting key information from a context that contains misleading items. The difference is that the context for Single3 consists of natural text, where the key information is a string of random alphanumeric characters, whereas MultiKey3 features a stack of key-value pairs composed of those strings. We find that while all three methods can succeed in the Single3 test, only full-attention-based Dynamic NTK can effectively handle MultiKey3.

To uncover the mechanisms behind the observation, we perform a t-SNE visualization (Van der Maaten & Hinton, 2008) of the K cache in LLaMA3-8B-8K using Dynamic NTK for both Single3 and MultiKey3, using the K cache in the last layer as an example. The results are shown in Figure 8. For Single3, which primarily consists of natural text, the reduced K cache exhibits a single manifold

|            | S3     | MK3   | All   |
|------------|--------|-------|-------|
| *8K Context Length* |    |       |       |
| DynamicNTK | 100.00 | 98.00 | 91.47 |
| InfLLM     | 45.00  | 38.00 | 44.91 |
| ReAttention | 95.00 | 78.00 | 90.10 |
| *16K Context Length* |   |       |       |
| DynamicNTK | 91.00  | 95.00 | 88.13 |
| InfLLM     | 18.00  | 17.00 | 29.61 |
| ReAttention | 94.00 | 15.00 | 66.57 |

Table 4: The performance of Dynamic NTK, InfLLM, and ReAttention on RULER benchmark in 8K and 16K context length. S3 and MK3 are the short forms of NIAH-Single3 and NIAH-Multikey3 respectively.

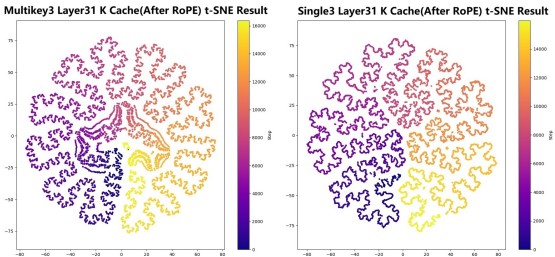

Figure 8: Visualization of the K Cache from the final layer of LLaMA3-8B-8K with Dynamic NTK extrapolation after inputs randomly sampled from the Single3 and MultiKey3 subsets in RULER. The visualization uses a 2D t-SNE projection, with each token represented as a point in the image and the input index shown via the color changing.

with a local cluster and an overall coherent feature as the input length increases. In contrast, for MultiKey3, which is dominated by chaotic text (Lv et al., 2024), the reduced K cache reveals multiple overlapping manifolds rather than an extension along a single curve. This results in the fact that during the K cache selection in MultiKey3, regardless of whether we use the same index encoding or direct sequential encoding for the selected segments, the subsequent self-attention fails to identify which manifold branch the segments belong to. Only by encoding all K caches according to their original position indices can LLM implicitly recognize (Chi et al., 2023; Kazemnejad et al., 2024) which manifold branch it resides on. This is why only full attention can effectively handle such tasks.

Nevertheless, the synthetic chaotic long texts are quite rare in practical scenarios (Lv et al., 2024). In the common natural long text benchmarks we have tested, ReAttention consistently demonstrates strong performance. Therefore, ReAttention still remains competitive for application in extending the context length of LLMs to infinity. We provide an in-depth discussion of the characteristics reflected in the QK sequence extension patterns between synthetic and natural texts in Appendix E.

## 5 RELATED WORK

Length extrapolation is an important issue for LLMs (Press et al., 2022), namely training in a short context, and maintaining good performance in a longer context. The mainstream extrapolation research mainly focuses on adjusting the Rotary Position Embedding (RoPE) (Su et al., 2021). For example, Linear PI (Chen et al., 2023) first achieves the length extrapolation in LLMs by scaling the position indices to the pre-training range with little fine-tuning. The NTK method (bloc97, 2023b;a; Peng et al., 2023) then adjusts the rotary base in RoPE (Su et al., 2021) to achieve plug-and-play length extrapolation. Subsequently, amplifying the rotary base and training on longer lengths has become the dominant approach for length extrapolation (Rozière et al., 2023; Xiong et al., 2023; Liu et al., 2023; Ding et al., 2024), but these methods all have an explicit extrapolation upper bound. In addition, ReRoPE (Su, 2023), Self-Extend (Jin et al., 2024), and ChunkLLaMA (An et al., 2024) also achieve plug-and-play extrapolation by limiting the relative position. However, all of the above methods are based on full attention, facing the problem of attention entropy soaring with input length, and thus fail to achieve infinite context length (Han et al., 2023; Wang et al., 2024).

In contrast, another line of research has tried to extend the context length of models through sparse attention. Considering the self-attention distribution tends to focus on the global and local context, StreamingLLM and LM-Infinite proposed the $\Lambda$-shaped attention window to achieve almost unlimited input length (Xiao et al., 2023; Han et al., 2023). However, since the input is accompanied by discarding the previous context, it still fails to extend the context length (Dong et al., 2024). Based on StreamingLLM, InfLLM (Xiao et al., 2024a) and LongHeads (Lu et al., 2024) attempt to extend context through chunkwise retrieval from the middle cache, but the semantic fragmentation and block representation issues also affect downstream task performance (Luo et al., 2024). Recently, MInference (Jiang et al., 2024) and RetrievalAttentionLiu et al. (2024a) utilize a dynamic cache selection to achieve significant speedup, but not attempt context extrapolation. In contrast, we propose ReAttention, which regards the cache selection as the preceding attention before the ordinary self-attention and achieves an infinite context with a finite attention scope, making LLMs free from the challenge of length extrapolation. We also minimize the top-$k$ attention kernel-wise like FlashAttention in self-attention (Dao et al., 2022; Dao, 2023). Additionally, since ReAttention is conducted based on cache selection, we also expand our discussion on cache optimization, especially on token eviction methods like H2O (Zhang et al., 2024d) and SnapKV, in Appendix D

## 6 CONCLUSION

In this paper, we introduce ReAttention, using a finite attention window to realize an infinite context length in each inference step. We evaluate ReAttention is on par with full attention in performance with LongBench, L-Eval, and InfiniteBench. Furthermore, ReAttention has been shown to successfully extend the context of mainstream LLMs, including the LLaMA series and Mistral, to up to 1M tokens and even expand the context length of LLaMA3.2-3B-chat by $128\times$ to 4M without any further training in Needle-In-A-Haystack tests. We also improve the efficiency of ReAttention with Triton and achieve an efficient extrapolation without additional overhead.

ACKNOWLEDGEMENT

This work is supported by the National Key Research and Development Program of China (No. U24B20181). We also appreciate the constructive comments from reviewers in the rebuttal and add discussion on methodology, effectiveness, efficiency, and related work in Appendix A to Appendix D.

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

## A    PSEUDOCODE OF REATTENTION

In this section, we present the pseudocode for ReAttention's prefilling and decoding phases. Our approach uses position-agnostic cache selection at each generation step, as shown in Section 2.1.

## B    MORE VALIDATION ON EFFECTIVENESS

Table 5 includes a detailed comparison of various LLMs including LLaMA (Meta, 2024a; Dubey et al., 2024; Meta, 2024b), Mistral(mistralai, 2024), InternLM (Cai et al., 2024b; InternLM, 2024), and Qwen (Yang et al., 2024a) series in LongBench (Bai et al., 2023) and L-Eval (An et al., 2023).

---

**Algorithm 1:** Prefilling Phase

---

**Input:** The input token indices input_ids, number of tokens $L$, the number of starting tokens $l_{\text{global}}$, the number of ending tokens $l_{\text{local}}$, size of prefilling chunk $l_{\text{chunk}}$.

1  **Initialize:** $s = 0$, $e = l_{\text{global}} + l_{\text{local}}$, past_key_values = None
2  **while** $s < L$ **do**
3     ▷ Process next chunk
4     past_key_values = None
5     input_chunk = input_ids[:, $s$:min($e$, $L$)]
6     hidden_states = `embedding`(input_chunk)
7     ▷ Forward through each transformer layer
8     **for** $i \leftarrow 0$ **to** $N_{layer} - 1$ **do**
9         ▷ Project to Q, K, V
10        query = `q_proj`(hidden_states)
11        key = `k_proj`(hidden_states)
12        value = `v_proj`(hidden_states)
13        ▷ Update KV cache for this layer
14        **if** past_key_values **then**
15           key = `concat`(past_key_values[$i$].key, key)
16           value = `concat`(past_key_values[$i$].value, value)
17        past_key_values[$i$] = (key, value)
18        ▷ Position-agnostic cache selection
19        score = query @ key[$l_{\text{global}}$:-$l_{\text{local}}$]$^T$
20        idx_recall = `topk`(score, dim=0, k=$k$)
21        key_recall = `concat`(
22           past_key_values[$i$].key[$l_{\text{global}}$:],
23           past_key_values[$i$].key[idx_recall],
24           past_key_values[$i$].key[:-$l_{\text{local}}$]
25        )
26        value_recall = `concat`(
27           past_key_values[$i$].value[$l_{\text{global}}$:],
28           past_key_values[$i$].value[idx_recall],
29           past_key_values[$i$].value[:-$l_{\text{local}}$]
30        )
31        ▷ Compute self-attention with position embedding
32        attention_output = `rope_attention`(query, key_recall, value_recall)
33        hidden_states = `ffn`(attention_output)
34     ▷ Move to the next chunk
35     $s = e$
36     $e = e + l_{\text{chunk}}$
37  **return** hidden_states, past_key_values

---

## C   MORE COMPARISON ON EFFICIENCY

Regarding the comparison between full attention and ReAttention in efficiency, the throughput and TTFT (Time To First Token) are two different measuring dimensions. TTFT measures performance under identical conditions (same batch size and context length). As shown in Table 6, our TTFT results demonstrate that ReAttention achieves latency on par with full attention equipped with FlashAttention2 under these controlled conditions. Using a single A800 GPU, we test our ReAttention approach and the standard implementation of Hugging Face's Llama3-8B (Meta, 2024b) modeling with FlashAttention2 (Dao, 2023). We keep all the settings unchanged and set the batch size to 1.

Throughput, on the other hand, evaluates maximum processing capacity when fully utilizing available GPU memory. As shown in Table 7, ReAttention demonstrates higher efficiency by supporting larger batch sizes while maintaining comparable or better token processing rates. Besides, beyond context length 64K, full attention equipped with FlashAttention2 (Dao, 2023) encounters memory limitations, while ReAttention continues working. Using a single A800 GPU, we test our ReAttention approach

---

**Algorithm 2:** Decoding Phase

---

**Input:** KV cache past_key_values, the last token index last_token, number of tokens $L$, the number of starting tokens $l_{\text{global}}$, the number of ending tokens $l_{\text{local}}$.

1  hidden_states = `embedding`(last_token)
2  ▷ Forward through each transformer layer
3  **for** $i \leftarrow 0$ **to** $N_{layer} - 1$ **do**
4     ▷ Project to Q, K, V
5     query = `q_proj`(hidden_states)
6     key = `k_proj`(hidden_states)
7     value = `v_proj`(hidden_states)
8     ▷ Update KV cache for this layer
9     key = `concat`(past_key_values[$i$].key, key)
10    value = `concat`(past_key_values[$i$].value, value)
11    past_key_values[$i$] = (key, value)
12    ▷ Position-agnostic cache selection
13    score = query @ key[$l_{\text{global}}$:-$l_{\text{local}}$]$^T$
14    idx_recall = `topk`(score, dim=0, k=$k$)
15    key_recall = `concat`(
16        past_key_values[$i$].key[$l_{\text{global}}$:],
17        past_key_values[$i$].key[idx_recall],
18        past_key_values[$i$].key[:-$l_{\text{local}}$]
19    )
20    value_recall = `concat`(
21        past_key_values[$i$].value[$l_{\text{global}}$:],
22        past_key_values[$i$].value[idx_recall],
23        past_key_values[$i$].value[:-$l_{\text{local}}$]
24    )
25    ▷ Compute self-attention with position embedding
26    attention_output = `rope_attention`(query, key_recall, value_recall)
27    hidden_states = `ffn`(attention_output)
28  ▷ Get the next token prediction
29  logits = `lm_head`(hidden_states)
30  next_token = `argmax`(logits)
31  **return** next_token, past_key_values

---

and full attention with FlashAttention2 (Dao, 2023) with LLaMA3-8B (Meta, 2024b). We keep all the settings unchanged. The generation length is 1024. This performance pattern indicates not a trade-off where we sacrifice one capability for another. Rather, it represents extending context length while maintaining competitive throughput and enabling larger batch sizes.

# D  More Related Work

The evolution of token eviction techniques begins with DCP (Anagnostidis et al., 2024), Scissorhands (Liu et al., 2024b), and H2O (Zhang et al., 2024d). These approaches demonstrate that retaining only a small subset of tokens with high attention scores could maintain near-equivalent performance while significantly reducing memory and computational overhead. Then, StreamingLLM (Xiao et al., 2023) advances this concept by identifying that critical tokens always occur in the beginning and end part of the attention scope. By preserving only corresponding tokens, StreamingLLM enables stable model outputs after infinite input lengths, dramatically reducing computational and memory costs. However, this approach substantially compromised long-context performance (Li et al., 2025; Xiao et al., 2024b). Subsequent researches explore mitigation strategies, such as head-adaptive methods like FastGen (Ge et al., 2023) and DuoAttention (Xiao et al., 2024b). Specifically, SnapKV (Li et al., 2025) introduced a more nuanced approach to pruning previous KV caches by querying the above context with later input segments, substantially enhancing long-context performance for token eviction techniques. Building upon these foundations, researchers such as PyramidKV (Cai et al., 2024a), PyramidInfer (Yang et al., 2024b), Gemfilter (Shi et al., 2024), and LongGen (Ge et al.,

| | LongBench | | | | | | | L-Eval | | |
|---|---|---|---|---|---|---|---|---|---|---|
| | S-Doc | M-Doc | Sum | ICL | Syn. | Code | Avg. | Close-Ended | Open-Ended | Avg. |
| *LLaMA3-8B-8K* | 18.08 | 13.17 | 21.42 | 63.95 | 10.45 | 69.89 | **35.48** | 46.11 | 2.79 | **17.23** |
| + StreamingLLM | 15.14 | 8.90 | 18.36 | 58.19 | 7.28 | 70.27 | 32.49 | 36.52 | 1.36 | 13.08 |
| + ReAttention | 19.08 | 11.33 | 20.69 | 63.40 | 8.95 | 70.38 | 35.03 | 44.94 | 2.43 | 16.60 |
| *LLaMA3.1-8B-128K* | 23.61 | 15.29 | 16.15 | 64.61 | 31.81 | 70.63 | **38.79** | 45.30 | 0.50 | **15.44** |
| + StreamingLLM | 21.67 | 11.46 | 14.01 | 61.05 | 33.43 | 70.50 | 37.06 | 42.36 | 0.38 | 14.37 |
| + ReAttention | 23.97 | 15.22 | 15.66 | 64.54 | 31.20 | 70.48 | 38.63 | 45.25 | 0.50 | 15.42 |
| *LLaMA3.2-3B-128K* | 34.45 | 26.24 | 24.46 | 59.29 | 40.11 | 55.73 | **40.76** | 55.75 | 20.63 | **32.33** |
| + StreamingLLM | 28.05 | 19.41 | 22.67 | 55.23 | 22.63 | 54.16 | 35.08 | 43.27 | 18.66 | 26.87 |
| + ReAttention | 32.53 | 24.39 | 24.13 | 58.13 | 37.39 | 55.17 | 39.43 | 53.09 | 20.12 | 31.11 |
| *LLaMA3.2-3B-128K* | 26.46 | 15.60 | 21.88 | 67.39 | 66.62 | 74.89 | 46.02 | 60.78 | 0.15 | 20.36 |
| + StreamingLLM | 23.72 | 12.14 | 20.66 | 65.35 | 56.36 | 75.06 | 43.20 | 55.32 | 0.16 | 18.54 |
| + ReAttention | 27.23 | 15.38 | 21.67 | 67.58 | 66.35 | 75.26 | **46.15** | 61.11 | 0.18 | **20.49** |
| *Mistral-v0.3-7B-32K* | 19.93 | 11.22 | 22.32 | 62.05 | 20.97 | 65.13 | **35.61** | 41.97 | 4.33 | **16.88** |
| + StreamingLLM | 17.60 | 9.79 | 20.68 | 59.32 | 19.42 | 64.85 | 34.01 | 36.49 | 4.45 | 15.13 |
| + ReAttention | 20.20 | 10.64 | 22.41 | 61.57 | 20.58 | 65.32 | 35.48 | 41.95 | 4.09 | 16.71 |
| *InternLM2.5-7B-1M* | 44.18 | 41.06 | 23.17 | 61.85 | 63.17 | 52.62 | 47.27 | 62.96 | 24.99 | **37.64** |
| + StreamingLLM | 41.26 | 34.46 | 22.26 | 59.58 | 53.67 | 52.97 | 44.03 | 57.97 | 23.30 | 34.85 |
| + ReAttention | 43.85 | 41.10 | 23.14 | 62.15 | 63.67 | 52.72 | **47.35** | 63.47 | 24.36 | 37.40 |
| *Qwen2-7B-128K* | 37.75 | 14.82 | 26.16 | 62.81 | 29.00 | 67.65 | **41.34** | 53.43 | 15.41 | **28.08** |
| + StreamingLLM | 35.46 | 11.22 | 25.11 | 61.46 | 23.00 | 67.39 | 39.16 | 50.07 | 14.30 | 26.22 |
| + ReAttention | 37.64 | 14.39 | 26.24 | 62.90 | 27.00 | 67.56 | 41.01 | 53.18 | 15.35 | 27.96 |
| *Qwen2-72B-128K* | 44.61 | 54.04 | 26.33 | 65.50 | 64.33 | 71.15 | **54.71** | 68.17 | 25.44 | **39.68** |
| + StreamingLLM | 40.11 | 40.24 | 25.29 | 64.48 | 54.83 | 70.91 | 50.08 | 60.43 | 22.38 | 35.06 |
| + ReAttention | 44.96 | 53.20 | 26.18 | 66.00 | 62.67 | 71.39 | 54.53 | 67.31 | 25.34 | 39.33 |
| *Qwen2-1B-32K* | 24.07 | 12.96 | 23.45 | 58.14 | 10.17 | 57.98 | 33.21 | 41.28 | 15.59 | 24.16 |
| + StreamingLLM | 23.87 | 10.13 | 22.87 | 56.62 | 10.00 | 58.32 | 32.40 | 37.46 | 15.31 | 22.69 |
| + ReAttention | 24.55 | 13.38 | 23.44 | 58.27 | 10.17 | 58.58 | **33.51** | 41.00 | 16.10 | **24.40** |

Table 5: Results of existing mainstream LLMs, including LLaMA Series and Mistral on LongBench and L-Eval. ReAttention achieves a consistent superiority over StreamingLLM and shows comparable performance with LLMs with full attention.

| | ReAttention (ms) | Baseline (ms) |
|---|---|---|
| 32k | 3984.66 | 3592.32 |
| 64k | 10696.22 | 9823.82 |
| 96k | 20139.42 | 18942.57 |
| 128k | 32328.20 | 31953.82 |
| 192k | 64832.10 | 64475.56 |
| 256k | 108240.09 | OOM |

Table 6: Comparison on TTFT.

| | ReAttention | | Baseline | |
|---|---|---|---|---|
| | token/s | batch | token/s | batch |
| 32k | 43.380 | 8 | 40.595 | 2 |
| 64k | 24.383 | 5 | 20.399 | 1 |
| 96k | 17.005 | 4 | OOM | |
| 128k | 11.683 | 2 | OOM | |
| 192k | 7.2813 | 1 | OOM | |
| 256k | 5.9680 | 1 | OOM | |

Table 7: Comparison on Throughput.

2024) take layer-wise optimization, maintaining inference effectiveness while improving memory efficiency further. Besides, works like VATP (Guo et al., 2024) and InfiniPot (Kim et al., 2024) expand token pruning metrics beyond traditional QK dot product scoring. Our ReAttention draws inspiration from these token eviction methods, specifically applying the principle of selecting KV cache to context extrapolation, enabling comprehensive context perception throughout the inference process while computationally addressing each attention calculation with a finite context, thereby achieving algorithmically infinite context length.

# E   ANALYSIS OF CACHE FEATURE

## E.1   THE METHOD FOR OBTAINING T-SNE RESULTS

We begin by extracting a lengthy sample (approximately 16K tokens) of text and feeding it into the LLaMA3-8B-8K (Meta, 2024a). Subsequently, we capture the $\mathcal{Q}, \mathcal{K}$ states at each layer, both before and after position embedding, allowing for a clearer understanding of the impact of position embedding on data representation. Next, the captured matrices are reshaped to (seqlen, nhead * headdim), and we apply t-SNE (Van der Maaten & Hinton, 2008) for dimension reduction, mapping the high-dimensional data into two dimensions with a final shape of (seqlen, 2).

## E.2   CACHE FEATURE OF LONG NATURAL LANGUAGE TEXTS

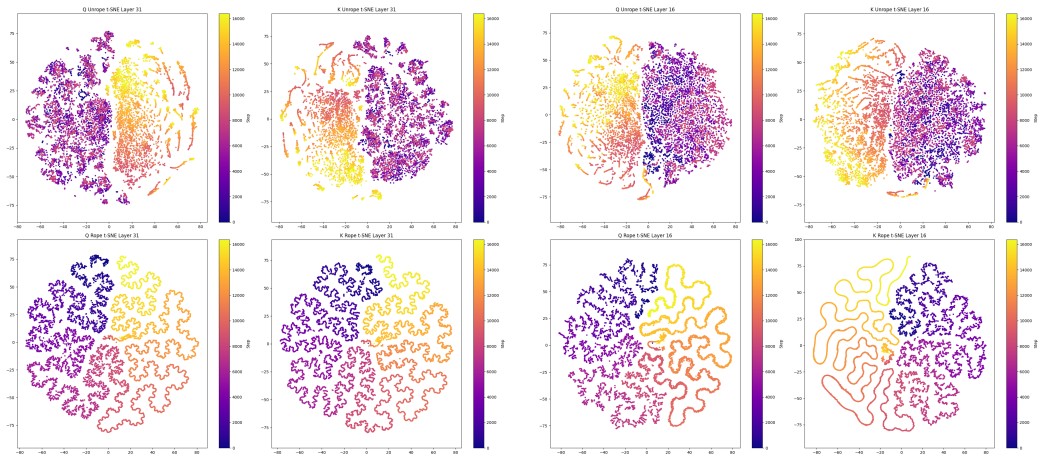

(a) t-SNE results of $\mathcal{Q}, \mathcal{K}$ from a data sample from the PG19 training set in the final layer of LLaMA3-8B-8K

(b) t-SNE results of $\mathcal{Q}, \mathcal{K}$ from a data sample from the PG19 training set in the 15[th] layer of LLaMA3-8B-8K

Figure 9: The 2-D reduction results of $\mathcal{Q}, \mathcal{K}$ from the final/15[th] layer, both before and after position embedding. Each figure is divided into four sections, representing four dimensionality reduction results. The left side corresponds to $\mathcal{Q}$, while the right side corresponds to $\mathcal{K}$. The top displays results before position embedding, and the bottom shows results after position embeddingg. Each point represents a token. The colorbar indicates token indices(positions), with purple representing the beginning of the sentence and yellow representing the end.

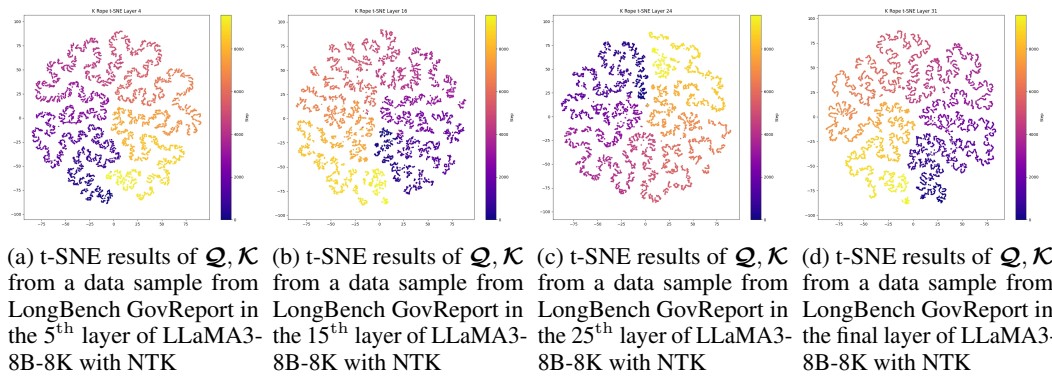

(a) t-SNE results of $\mathcal{Q}, \mathcal{K}$ from a data sample from LongBench GovReport in the 5[th] layer of LLaMA3-8B-8K with NTK

(b) t-SNE results of $\mathcal{Q}, \mathcal{K}$ from a data sample from LongBench GovReport in the 15[th] layer of LLaMA3-8B-8K with NTK

(c) t-SNE results of $\mathcal{Q}, \mathcal{K}$ from a data sample from LongBench GovReport in the 25[th] layer of LLaMA3-8B-8K with NTK

(d) t-SNE results of $\mathcal{Q}, \mathcal{K}$ from a data sample from LongBench GovReport in the final layer of LLaMA3-8B-8K with NTK

Figure 10: Given the smaller number of tokens of this sample, the clusters are distinctly separated by the algorithm. Despite some dispersion in the intermediate layers, the data representation maintains alignment along a linear manifold in the higher layers, indicating the robustness of the feature of Global Coherence.

The $\mathcal{Q}, \mathcal{K}$ states of long natural language text preserves coherence, which can be visualized as a curve extending through high-dimensional space, encompassing the entire text. When the model extracts contextual information, it effectively accesses specific points along this line. The visualization in Figure 9 and Figure 10 reveals several key features:

- Local Semantic Similarity: Locally, the semantic meaning of individual text segments is preserved, forming small clusters. (Zandieh et al., 2024)

- Global Coherence: These local clusters, however, are not isolated but part of a larger linear manifold representing the text's global coherence. This coherence becomes more apparent after position embedding is applied. (Zandieh et al., 2024)

- Out-of-Distribution Position Embedding: When the position embedding exceeds 8K, Llama3-8B struggles to extract semantic information, as seen in the diminished clustering. (Liu et al., 2023) Nonetheless, even when the position embedding is out-of-distribution(OOD), the text representation continues to align with a linear manifold.

### E.3 FROM NATURAL LANGUAGE TEXTS TO CHAOTIC TEXTS

In contrast, Ruler's MultiKey3 subset contains unnaturally long texts that deviate from typical linguistic patterns, lacking the structure and semantic coherence of natural language. Here in this subset, the text length is artificially extended to 16k tokens, not by model generation but by repeating specific patterns

> One of the special magic UUIDs for A is: B.

like

> One of the special magic UUIDs for eab8d5be-b62c-49bf-a4f7-3a3017a15a60
> is: 960e9c1a-b4c0-4d18-a5cd-7fd162649ce8.

Moreover, large portions of these patterns consist of non-linguistic, non-logical elements like UUIDs.

When applying t-SNE to reduce the dimension of MultiKey3 data(Figure 11), the linear manifold observed in natural language text vanishes, highlighting the impact of unnatural patterns on data representation. Key observations include:

- Local Semantic Similarity: By examining $\mathcal{Q}$ after position embedding, we can clearly see that the UUIDs and natural language components are completely separated, even in the deep purple section at the start. In $\mathcal{K}$ after position embedding, adjacent tokens (i.e., data points of similar colors) are dispersed across multiple lines, thus making the term 'local similarity' questionable.

- Global Coherence: The assumption that the entire data representation lies on a linear manifold no longer holds. Unnaturally extended texts, unlike typical natural language, fail to maintain global structural dependencies.

The effectiveness of dynamic sparse methods in replacing the full attention mechanism hinges on the collaboration between global and local information. However, when text coherence is lacking, even adjacent tokens in the $\mathcal{Q}, \mathcal{K}$ cache become dispersed, disrupting the required linear manifold structure that LLM relies on. This suggests that LLM focuses its top-$k$ attention on natural language segments and successfully extracts critical information. Yet, when the top-$k$ values are spread across multiple manifolds, LLM fails to utilize this information efficiently. This contrasts sharply with the typical feature of the $\mathcal{Q}, \mathcal{K}$ cache in natural language, where the representation forms a continuous, intertwined structure across the text in high-dimensional space. It is precisely the absence of this characteristic feature that causes the cache generated during prefilling with ReAttention to be biased when processing RULER Multikey3, leading to the output of non-existent UUIDs.

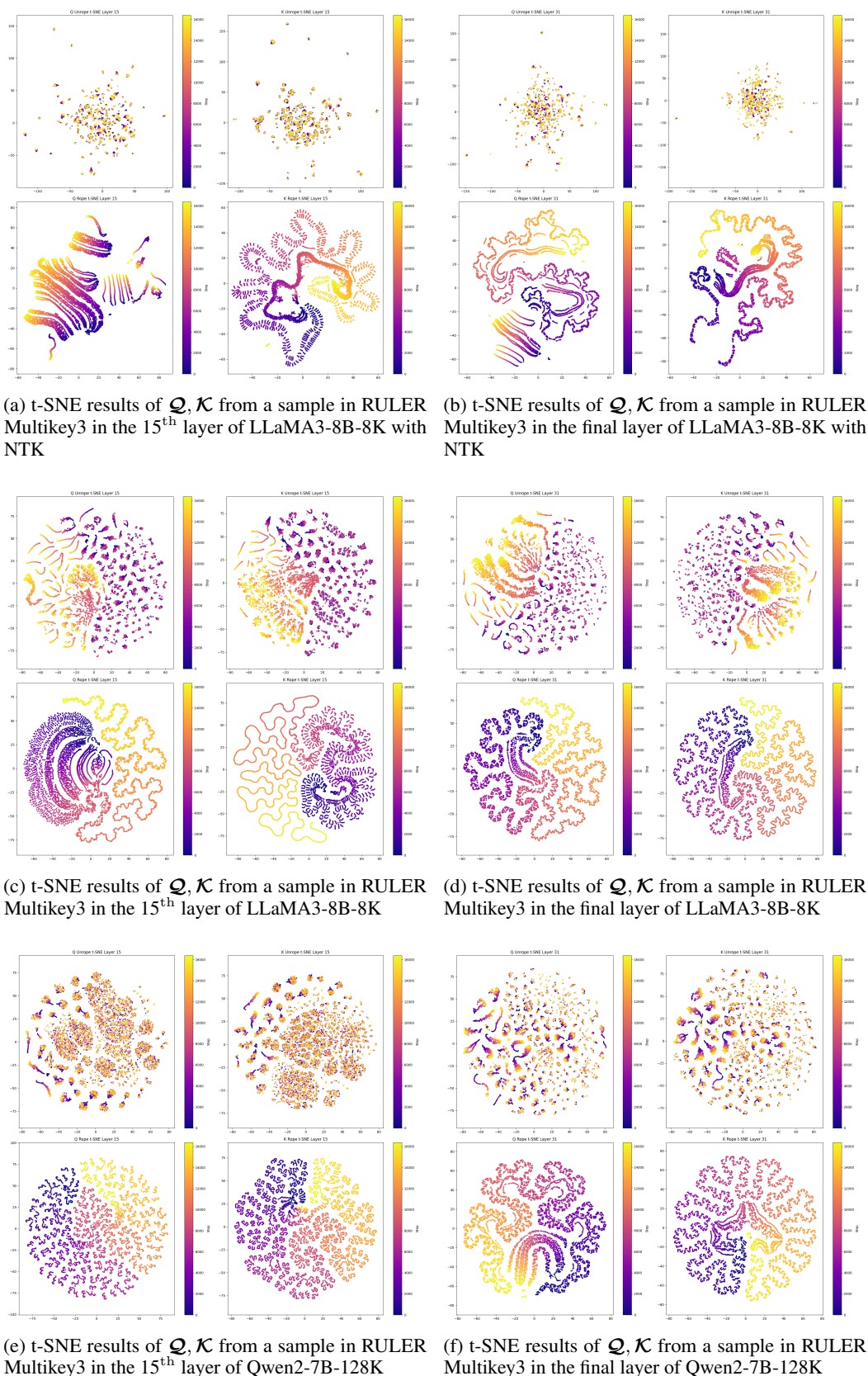

(a) t-SNE results of $\mathcal{Q}, \mathcal{K}$ from a sample in RULER Multikey3 in the $15^{\text{th}}$ layer of LLaMA3-8B-8K with NTK

(b) t-SNE results of $\mathcal{Q}, \mathcal{K}$ from a sample in RULER Multikey3 in the final layer of LLaMA3-8B-8K with NTK

(c) t-SNE results of $\mathcal{Q}, \mathcal{K}$ from a sample in RULER Multikey3 in the $15^{\text{th}}$ layer of LLaMA3-8B-8K

(d) t-SNE results of $\mathcal{Q}, \mathcal{K}$ from a sample in RULER Multikey3 in the final layer of LLaMA3-8B-8K

(e) t-SNE results of $\mathcal{Q}, \mathcal{K}$ from a sample in RULER Multikey3 in the $15^{\text{th}}$ layer of Qwen2-7B-128K

(f) t-SNE results of $\mathcal{Q}, \mathcal{K}$ from a sample in RULER Multikey3 in the final layer of Qwen2-7B-128K

Figure 11: t-SNE visualization results of different models under RULER Multikey3.

