# OpenReview forum: "ReAttention: Training-Free Infinite Context with Finite Attention Scope"
_ICLR.cc/2025/Conference — ICLR 2025 Poster_

### Official Review · Reviewer_bDJC · 2024-11-01

**Soundness:** 2
**Presentation:** 2
**Contribution:** 2
**Rating:** 5
**Confidence:** 5

**Summary:**

The paper introduces ReAttention, a training-free approach for enabling large language models (LLMs) to handle infinite context lengths with a finite attention scope. The method involves a position-agnostic top-k attention mechanism, which precedes traditional self-attention, to select the most relevant context segments. This approach addresses the limitations of context length in LLMs without the need for additional training. The authors validate ReAttention across multiple benchmarks, demonstrating its effectiveness and efficiency compared to existing methods.

**Strengths:**

1. The use of Triton to optimize the attention mechanism reduces computational overhead and memory usage.
2. The method supports significant context length extensions, showcasing its potential for real-world applications.

**Weaknesses:**

1. The paper uses the top-K method to select KV, but this has been studied in H2O. It feels like the paper is a combination of StreamingLLM+H2O, and then rewrites Triton to make up for the calculation time of QK. I don't see any novelty. It would be helpful to provide more specific details on how ReAttention differs from or combines elements of StreamingLLM and H2O.

2. The rewrite of the Triton operator is indeed very effective, but it is more of an engineering contribution.

3. The paper is mainly an improvement on KV. There are so many optimizations on KV, but there are so few baselines added for comparison. It is recommended to select some baselines from SnapKV, PyramidKV, LongGen, InfiniPot, DuoAttention and other methods for comparison.

4. Many experiments in the paper show good results, but there is no intuitive or formal explanation for why this method can achieve such good results. For example, theoretical analysis, or visualizations that might help explain the performance gains.

**Questions:**

see weaknesses

---

> ### Author Response · Authors · 2024-11-18
> **Response (1/2)**
>
> Dear Reviewer bDJC
>
> We thank you for your valuable feedback and have addressed your concerns as follows.
>
> Regarding W1, on the similarity with other works, we do have similarities with StreamingLLM [1] and H2O [2], in that we preserve the beginning and end tokens during attention and perform cache selection from the middle. However, ReAttention is fundamentally different from StreamingLLM and H2O. ReAttention is a work on extending context length, not a KV cache optimization method[1-2]. If preserving the beginning and end tokens and filtering middle tokens is concerned, there are many works including InfLLM (accepted by NeurIPS 2024) [3], LongHeads (accepted by EMNLP 2024) [4], TOVA [5], and additionally the works mentioned by the reviewer such as SnapKV (accepted by NeurIPS 2024) [6], PyramidKV [7], DuoAttention [8], which also preserve adjacent tokens and filter previous tokens.
>
> Regarding W3, on comparisons with KV cache optimization works, first, we need to clarify that we are not a KV cache optimization work, but a work on extending model context length. Among the KV cache optimization works mentioned by the reviewer, SnapKV observes attention feature differences across different heads and clips historical tokens based on tokens at the prompt's end [6]. PyramidKV observes sparsity of attention patterns between different layers, and based on SnapKV, sets different numbers of KV cache storage in different layers to further improve computational and storage efficiency [7]. These works achieve efficient inference in a plug-and-play manner but cannot effectively achieve extrapolation. DuoAttention, building on SnapKV's approach, trains different attention heads' tendencies towards StreamingLLM and full attention, artificially classifying attention heads and applying different attention patterns during inference [8]. LongGen fine-tunes the model's intermediate layers to adapt to sparse attention patterns while maintaining full attention in the first and last layers [9]. Compared to these methods, our approach can effectively extrapolate without any training. InfiniPot is also a KV cache optimization work, defining importance metrics from future and past tokens as new token clipping criteria [10]; we similarly discussed cache selection in our paper for better context extrapolation.
>
> Considering the reviewer's concern about ReAttention's downstream task performance, we choose the more influential SnapKV for downstream task comparisons with LongBench [11] and find that ReAttention still demonstrates competitive performance.

---

> ### Author Response · Authors · 2024-11-18
> **Response (2/2)**
>
> Results on LongBench. All experiments are conducted in 32k context length, while LLaMA3-8B-8K is evaluated with Dynamic NTK.
>
> | | En | Zh | Code | Avg. |
> |-----|:-----:|:-----:|:-----:|:-----:|
> | LLaMA3-8B-8K | 34.27 | 19.95 | 69.89 | 35.48 |
> | + StreamingLLM | 29.48 | 13.22 | 70.27 | 32.49 |
> | + InfLLM | 29.53 | 17.60 | 70.03 | 32.51 |
> | + SnapKV | 29.70 | 16.36 | 69.41 | 33.09 |
> | + ReAttention | 33.42 | 18.74 | 70.38 | 35.03 |
> | LLaMA3.1-8B-128K | 36.38 | 28.64 | 70.63 | 38.79 |
> | + StreamingLLM | 33.41 | 24.24 | 70.50 | 37.06 |
> | + InfLLM | 34.26 | 27.22 | 68.37 | 37.14 |
> | + SnapKV | 34.25 | 25.22 | 70.60 | 37.73 |
> | + ReAttention | 36.14 | 28.56 | 70.48 | 38.63 |
>
> Regarding W4, on theoretical analysis and visualization explanations, first, we thank the reviewer for acknowledging our experimental results. Regarding the latter, we have detailed descriptions in our paper, refer to subsection 4.2. Through experimental observations, we discover that token filtering without positional embeddings can more effectively locate key information in the previous context, enabling ReAttention to solve problems that full attention cannot. Regarding theoretical analysis, we are actively discussing this and will treat it as our future work.
>
> [1] Xiao, Guangxuan, et al. "Efficient Streaming Language Models with Attention Sinks." The Twelfth International Conference on Learning Representations.
>
> [2] Zhang, Zhenyu, et al. "H2O: Heavy-hitter oracle for efficient generative inference of large language models." Advances in Neural Information Processing Systems 36 (2023): 34661-34710.
>
> [3] Xiao, Chaojun, et al. "Infllm: Training-free long-context extrapolation for llms with an efficient context memory." The Thirty-eighth Annual Conference on Neural Information Processing Systems. 2024.
>
> [4] Lu, Yi, et al. "LongHeads: Multi-Head Attention is Secretly a Long Context Processor." arXiv preprint arXiv:2402.10685 (2024).
>
> [5] Oren, Matanel, et al. Transformers are multi-state RNNs. arXiv preprint arXiv:2401.06104 (2024).
>
> [6] Li, Yuhong, et al. "SnapKV: LLM knows what you are looking for before generation." arXiv preprint arXiv:2404.14469 (2024).
>
> [7] Cai, Zefan, et al. "PyramidKV: Dynamic kv cache compression based on pyramidal information funneling." arXiv preprint arXiv:2406.02069 (2024).
>
> [8] Xiao, Guangxuan, et al. "DuoAttention: Efficient Long-Context LLM Inference with Retrieval and Streaming Heads." arXiv preprint arXiv:2410.10819 (2024).
>
> [9] Ge, Suyu, et al. "A little goes a long way: Efficient long context training and inference with partial contexts." arXiv preprint arXiv:2410.01485 (2024).
>
> [10] Kim, Minsoo, et al. "InfiniPot: Infinite Context Processing on Memory-Constrained LLMs." Proceedings of the 2024 Conference on Empirical Methods in Natural Language Processing. 2024.
>
> [11] Bai, Yushi, et al. Longbench: A bilingual, multitask benchmark for long context understanding. arXiv preprint arXiv:2308.14508 (2023).

---

> > ### Comment · Reviewer_bDJC · 2024-11-26
> >
> > Thanks for the author's detailed reply.
> > 1. It is true that many works involving preserving the start and end tokens and filtering the middle tokens have been accepted, but as you said, they all have their own innovations. Based on your discussion with other reviewers, I understand that your innovation is position-independent. Therefore, I will increase the score to 5 points.
> > 2. I don't agree with what you said that "ReAttention is fundamentally different from StreamingLLM and H2O". If this is the case, then why do you still need to compare with StreamingLLM. Since you compared, why don't you compare with the recent SOTA. Although you said that you are a work that extends the context length of the model, your method is KV cache optimization. I don't think your answer can convince me. Therefore, I will no longer increase the score.

---

> > > ### Author Response · Authors · 2024-11-26
> > > **Response**
> > >
> > > Dear Reviewer,
> > >
> > > We sincerely appreciate your continued engagement and questions regarding our work. First of all, I would like to thank you for your recognition of our innovation. Regarding the difference between our method and StreamingLLM, we are committed to more detailed comparisons and more explicit methodology explanations, ensuring transparent and comprehensive communication of our technical contributions.
> > >
> > > First, while StreamingLLM primarily focuses on KV cache optimization through drop token in the middle, ReAttention does not engage in token eviction or compression. More importantly, ReAttention not only extends the input length and ensures stable outputs in long-context scenarios like StreamingLLM, but also genuinely enables LLMs with longer context length to realize long-context understanding and retrieval. We have validated these claims through evaluations including LongBench, LEval, InfiniteBench, and NIAH benchmarks. We also compare ReAttention with the state-of-the-art plug-and-play context extension methods, like InfLLM, to further substantiate our approach's effectiveness.
> > >
> > > We remain open to further constructive dialogue and appreciate the opportunity to refine our work.
> > >
> > > Best wishes,
> > >
> > > The Authors

---

> ### Author Response · Authors · 2024-11-24
> **Looking forward to receiving your feedback**
>
> Dear Reviewer bDJC,
>
> We hope we have adequately addressed your issues. We would be very grateful if you could provide feedback on our rebuttal since the deadline is approaching. If you require further clarification or have any additional concerns, please do not hesitate to contact us. We are more than willing to continue communicating with you.
>
> Best wishes,
>
> The Authors

---

### Official Review · Reviewer_fRQF · 2024-11-04

**Soundness:** 2
**Presentation:** 2
**Contribution:** 2
**Rating:** 6
**Confidence:** 3

**Summary:**

This paper proposes a training-free method (ReAttention) for improving the ability of pre-trained language models to process long context lengths. The method is motivated by the insight that attention scores without positional embeddings are still useful for identifying salient tokens.

The main claims of the paper are that:

1. ReAttention matches or outperforms the performance of traditional self-attention in long-context scenarios.
2. ReAttention matches or outperforms the performance of other training-free methods long-context methods (*e.g.* InfLLM).
3. ReAttention extends the context length of existing LLMs (*e.g.* 8B and 3B llama) “to at least 1 million tokens”.
4. ReAttention adds no computational overhead and less memory usage than regular self-attention.

**Strengths:**

*The method is motivated by an interesting observation.* “*We also find that the third condition can be satisfied through the attention score without positional embedding.” This insight that attention scores without positional embeddings can effectively identify salient tokens is quite interesting and, to my knowledge, novel.

*Method works on pre-trained models.* Unlike approaches like Mamba which require training models from scratch with new architectures, ReAttention can be applied directly to existing pre-trained models. This makes it much more practical for immediate deployment and use with current state-of-the-art models.

*Claim 1 seems to be well-supported by the results.*   The empirical evaluation in Figure 2 and Table 1  across multiple benchmarks (LongBench and InfiniteBench) and model architectures shows that ReAttention matches or outperforms regular attention.
- Note: in the weaknesses, I highlight some missing baselines.

**Weaknesses:**

*Baselines in the long context evals are missing (Claim 2). W*hy is StreamingLLM included in the LongBench results but not the InfiniteBench results? And why is InfLLM included in the InffiniteBench results but not the LongBench results?

*Baselines in million-scale context seem to be missing. (Claim 3)* For the needle in a haystack experiments supporting Claim 3, the paper should compare against other methods capable of handling very long contexts like InfLLM and other recent approaches. Without these comparisons, it's hard to assess the relative advantages of ReAttention at extreme context lengths.

*Missing token-throughput comparisons (Claim 3).* The paper only compares time to first token (*i.e.* the cost of prefill).  How does it fare against regular attention in terms of efficiency?

*Apparent tradeoff in memory consumption (Claim 4).* The authors state that ReAttention uses less memory than self-attention. However, based on Figure 7, it appears that this is only true after ~100k in context length. The authors should highlight this tradeoff.

*Method description lacks details.* It’s unclear exactly how the method works from the description in Section 2. For example, what does it mean that ReAttention “adopts chunked streaming input during the prefilling stage.” I recommend that the authors provide a detailed description of how prefill with ReAttention differs from decoding. It would be great if the authors included a pseudo-code description of both the prefill and decoding stages (either in the main paper or the appendix).

*The "infinite" in the title is misleading.* It’s not clear from the experiments that ReAttention actually supports infinite context. Further, there are still practical limitations from memory constraints that would prevent the model from really achieving infinite context length. The paper should be more precise about what "infinite" means in this context and acknowledge the real-world constraints.

**Questions:**

- *What is the performance of the baselines in the needle-in-a-haystack experiments.*
- *In Figure 7, is the FullAttn* implementation FlashAttention?
- *How exactly does ReAttention differ from InfLLM? Is the only difference that position embeddings are not applied during the lookup step?* (I recommend the authors update the paper to make this comparison more clear.)
- *Minor improvements:*
    - In Figures 3 and 4, the axes and color bars are too small to be legible.

---

> ### Author Response · Authors · 2024-11-18
> **Response (1/5)**
>
> Dear Reviewer fRQF
>
> We appreciate your valuable feedback and have addressed your concerns as follows.
>
> Regarding W1 on missing long-context evaluation results, thank you for highlighting the need for comprehensive evaluations. We have conducted additional experiments comparing ReAttention with StreamingLLM [1] on Infinitebench [2] and with InfLLM [3] on LongBench [4]. Our results demonstrate that ReAttention achieves the best performance on average.
>
> Results on InfiniteBench
>
> |   |   | 32k |   |   | 64k |   |   | 128k |   | Avg. |
> |-----|:-----:|:-----:|:-----:|:-----:|:-----:|:-----:|:-----:|:-----:|:-----:|:-----:|
> |   | **MC** | **QA** | **Sum** | **MC** | **QA** | **Sum** | **MC** | **QA** | **Sum** |   |
> | ***LLaMA3-8B-Instruct-8K*** | — | — | — | — | — | — | — | — | — | — |
> | + DynamicNTK | 50.66 | 7.44 | 21.76 | — | — | — | — | — | — | — |
> | + StreamingLLM | 27.95 | 4.40 | 18.41 | 28.38 | 4.30 | 17.61 | 34.93 | 4.36 | 17.64 | 17.55 |
> | + InfLLM | 40.61 | 4.15 | 20.30 | 35.37 | 5.12 | 18.26 | 33.19 | 5.78 | 19.70 | 20.28 |
> | + ReAttention | 46.29 | 5.33 | 21.76 | 44.54 | 5.25 | 20.15 | 49.78 | 5.98 | 20.36 | **24.38** |
> | ***LLaMA3.1-8B-Instruct-128K*** | 37.12 | 18.85 | 1.60 | 31.88 | 23.32 | 1.39 | 21.83 | 25.43 | 1.40 | 18.09 |
> | + StreamingLLM | 34.50 | 6.82 | 19.61 | 34.93 | 6.83 | 20.82 | 35.81 | 6.71 | 20.10 | 20.68 |
> | + InfLLM | 32.31 | 5.48 | 21.00 | 34.50 | 7.56 | 19.82 | 39.30 | 8.92 | 19.27 | 20.91 |
> | + ReAttention | 36.68 | 12.71 | 19.68 | 35.37 | 12.14 | 18.84 | 40.61 | 12.63 | 18.14 | **22.98** |
>
> Results on LongBench
>
> | | En | Zh | Code | Avg. |
> |-----|:-----:|:-----:|:-----:|:-----:|
> | ***LLaMA3-8B-8K*** | 33.91 | 18.98 | 69.80 | 35.25 |
> | + DynamicNTK | 34.27 | 19.95 | 69.89 | 35.48 |
> | + StreamingLLM | 29.48 | 13.22 | 70.27 | 32.49 |
> | + InfLLM | 29.53 | 17.60 | 70.03 | 32.51 |
> | + ReAttention | 33.42 | 18.74 | 70.38 | 35.03 |
> | ***LLaMA3.1-8B-128K*** | 36.38 | 28.64 | 70.63 | 38.79 |
> | + StreamingLLM | 33.41 | 24.24 | 70.50 | 37.06 |
> | + InfLLM | 34.26 | 27.22 | 68.37 | 37.14 |
> | + ReAttention | 36.14 | 28.56 | 70.48 |  38.63 |

---

> ### Author Response · Authors · 2024-11-18
> **Response (2/5)**
>
> Regarding W2 and Q1 on the NIAH comparison, we sincerely thank you for your suggestion. We have performed experiments comparing ReAttention with baseline methods and InfLLM [3] in LLaMA3-8B-8K. Our results conclusively show that ReAttention significantly outperforms FullAttention, FullAttention with DynamicNTK [5], and InfLLM on the NIAH task [6]. More results will be added to the paper later.
>
> LLaMA3-8B-8K with DynamicNTK, scaling_factor=4
>
> | depth \ length | 8k | 16k | 32k | 50k | 100k |
> |-----|:-----:|:-----:|:-----:|:-----:|:-----:|
> | 0% | 100.00 | 100.00 | 100.00 | 3.19 | OOM |
> | 10% | 100.00 | 100.00 | 100.00 | 3.23 | OOM |
> | 20% | 100.00 | 100.00 | 100.00 | 12.95 | OOM |
> | 30% | 100.00 | 100.00 | 100.00 | 3.20 | OOM |
> | 40% | 100.00 | 100.00 | 100.00 | 3.43 | OOM |
> | 50% | 100.00 | 100.00 | 100.00 | 3.28 | OOM |
> | 60% | 100.00 | 100.00 | 90.35 | 3.42 | OOM |
> | 70% | 100.00 | 100.00 | 100.00 | 3.46 | OOM |
> | 80% | 100.00 | 100.00 | 100.00 | 3.33 | OOM |
> | 90% | 100.00 | 100.00 | 100.00 | 3.52 | OOM |
> | 100% | 100.00 | 100.00 | 100.00 | 22.51 | OOM |
>
> LLaMA3-8B-8K with StreamingLLM, local_size=4096, global_size=32, chunk_size=512
>
> | depth \ length | 8k | 16k | 32k | 50k | 100k |
> |-----|:-----:|:-----:|:-----:|:-----:|:-----:|
> | 0% | 3.73 | 3.89 | 3.94 | 3.79 | 3.79 |
> | 10% | 3.73 | 3.85 | 3.94 | 3.79 | 3.79 |
> | 20% | 3.72 | 3.85 | 3.94 | 3.79 | 3.79 |
> | 30% | 3.73 | 3.85 | 3.94 | 3.80 | 3.79 |
> | 40% | 3.67 | 3.89 | 3.94 | 3.79 | 3.79 |
> | 50% | 100.00 | 3.85 | 3.94 | 3.79 | 3.79 |
> | 60% | 100.00 | 3.85 | 3.94 | 3.79 | 3.79 |
> | 70% | 100.00 | 100.00 | 3.94 | 3.79 | 3.79 |
> | 80% | 100.00 | 100.00 | 3.94 | 3.79 | 3.79 |
> | 90% | 100.00 | 100.00 | 100.00 | 3.79 | 3.79 |
> | 100% | 100.00 | 100.00 | 100.00 | 100.00 | 100.00 |
>
> LLaMA3-8B-8K with InfLLM, local_size=4096, global_size=128, seg_size=128 (recommended by InfLLM), num_seg=31, chunk_size=512
>
> | depth \ length | 8k | 16k | 32k | 50k | 100k |
> |-----|:-----:|:-----:|:-----:|:-----:|:-----:|
> | 0% | 61.78 | 32.79 | 22.34 | 22.57 | 61.18 |
> | 10% | 22.25 | 3.23 | 3.15 | 12.91 | 13.00 |
> | 20% | 31.96 | 12.59 | 13.03 | 32.75 | 41.92 |
> | 30% | 32.73 | 12.78 | 22.34 | 3.10 | 3.12 |
> | 40% | 22.51 | 12.73 | 12.82 | 3.57 | 3.38 |
> | 50% | 71.17 | 13.26 | 12.68 | 22.53 | 12.73 |
> | 60% | 90.32 | 12.66 | 12.55 | 12.90 | 13.06 |
> | 70% | 90.28 | 12.64 | 22.47 | 12.74 | 42.04 |
> | 80% | 90.39 | 90.35 | 22.53 | 12.99 | 32.34 |
> | 90% | 100.00 | 90.40 | 70.86 | 22.61 | 32.18 |
> | 100% | 100.00 | 100.00 | 100.00 | 100.00 | 100.00 |
>
> LLaMA3-8B-8K with ReAttention, local_size=4096, global_size=32, seg_size=32, num_seg=127=k', chunk_size=512
>
> | depth \ length | 8k | 16k | 32k | 50k | 100k | 200k | 300k | 400k |
> |-----|:-----:|:-----:|:-----:|:-----:|:-----:|:-----:|:-----:|:-----:|
> | 0% | 100.00 | 100.00 | 100.00 | 100.00 | 100.00 | 100.00 | 100.00 | 100.00 |
> | 10% | 100.00 | 100.00 | 100.00 | 100.00 | 100.00 | 90.42 | 100.00 | 100.00 |
> | 20% | 100.00 | 100.00 | 100.00 | 100.00 | 100.00 | 100.00 | 100.00 | 100.00 |
> | 30% | 100.00 | 100.00 | 100.00 | 100.00 | 100.00 | 100.00 | 100.00 | 100.00 |
> | 40% | 100.00 | 100.00 | 100.00 | 100.00 | 100.00 | 90.40 | 100.00 | 100.00 |
> | 50% | 100.00 | 100.00 | 100.00 | 100.00 | 100.00 | 100.00 | 100.00 | 100.00 |
> | 60% | 100.00 | 100.00 | 100.00 | 100.00 | 100.00 | 100.00 | 100.00 | 100.00 |
> | 70% | 100.00 | 100.00 | 100.00 | 100.00 | 100.00 | 100.00 | 100.00 | 100.00 |
> | 80% | 100.00 | 100.00 | 100.00 | 100.00 | 100.00 | 90.29 | 90.42 | 100.00 |
> | 90% | 100.00 | 100.00 | 100.00 | 100.00 | 100.00 | 90.42 | 100.00 | 100.00 |
> | 100% | 100.00 | 100.00 | 100.00 | 100.00 | 100.00 | 100.00 | 100.00 | 100.00 |

---

> ### Author Response · Authors · 2024-11-18
> **Response (3/5)**
>
> Regarding Q2 on Figure 7, we confirm that we use full attention with FlashAttention2 [7] as our baseline in the referenced figure.
>
> Regarding W3, we've now run a new set of experiments, looking at throughput during decoding on a single A800 GPU, a complement to the TTFT (Time To First Token) metrics presented in our paper. Using a single A800 GPU, we test our ReAttention approach and the standard implementation of Hugging Face's Llama 3 8B modeling with FlashAttention2. We keep all the settings, e.g. local size and global size, exactly the same as in our paper. The generation length is exactly 1024. Here's what we have found:
>
> | Context Length | Our Method (token/s) | Baseline (token/s) | batch size<br>(ours/baseline) |
> | -------------- | ----------------------- | -----------| ---- |
> | 32k | 43.380 | 40.595 | 8/2 |
> | 64k | 24.383 | 20.399 | 5/1 |
> | 96k | 17.005 | OOM | 4/- |
> | 128k | 11.683 | OOM | 2/- |
> | 192k | 7.2813 | OOM | 1/- |
> | 256k | 5.9680 | OOM | 1/- |
>
> Not only can we handle much longer sequences, but we're also faster at shorter lengths where both methods work. We're seeing about 7-20% better throughput at 32k and 64k contexts, while also managing larger batch sizes. Beyond that, the baseline runs out of memory, but our method keeps going strong while only using about 60% of the GPU memory(49.3GB at 256k).
>
> Regarding W4 on memory overhead tradeoff, we appreciate your insightful feedback on memory considerations. We are committed to continuous optimization of our Triton operator [8]. As a training-free context extrapolation method primarily, ReAttention focuses on demonstrating significant advantages in long-context scenarios. Specifically, our approach requires less memory compared to full attention in contexts exceeding 100k tokens and enables extrapolation of LLaMA3.1-8B to 400k tokens on a single GPU.

---

> ### Author Response · Authors · 2024-11-18
> **Response (4/5)**
>
> Regarding W5 in a detailed description, we appreciate the reviewer's concern.  Our approach uses dynamic token filtering at each generation step, which we covered in Section 2.1. We have prepared pseudocode for ReAttention's prefilling and decoding phases, which we plan to include in the appendix to avoid misunderstanding. Since the website's having some trouble rendering LaTeX right now, we've prepared a markdown version for you to look at.
>
> Prefilling Phase
>
> ```python
> # Input: seq_len tokens
> # Configuration:
> #   - global_size tokens at start of cache
> #   - local_size tokens at end of cache
> #   - chunk_size for processing
>
> start = 0
> end = global_size + local_size
> past_key_values = None
>
> while start < seq_len:
>     # Process next chunk
>     input_chunk = input_ids[:, start:min(end, seq_len)]
>     hidden_states = embedding(input_chunk)
>
>     # Forward through each transformer layer
>     for layer_i in range(num_layers):
>         # Project to Q,K,V
>         query = q_proj(hidden_states)
>         key = k_proj(hidden_states)
>         value = v_proj(hidden_states)
>
>         if layer_i == 0:
>             # Apply position embeddings
>             position_ids = get_position_ids(start, end)
>             cos, sin = get_rope_embeddings(position_ids)
>             query, key = apply_rope(query, key, cos, sin)
>
>         # Update KV cache for this layer
>         if past_key_values:
>             key = concat(past_key_values[layer_i].key, key)
>             value = concat(past_key_values[layer_i].value, value)
>         past_key_values[layer_i] = (key, value)
>
>         score = query @ (key[global_size:-local_size])^T
>               idx_recall = score.topk(dim=0)
>
>         key_recall = concat(past_key_values[layer_i].key[global_size:],
>                            past_key_values[layer_i].key[idx_recall],
>                            past_key_values[layer_i].key[:-local_size])
>         value_recall = concat(past_key_values[layer_i].value[global_size:],
>                            past_key_values[layer_i].value[idx_recall],
>                            past_key_values[layer_i].value[:-local_size])
>
>         # Compute attention
>         attention_output = flash_attention(query, key_recall, value_recall)
>         hidden_states = ffn(attention_output)
>
>     # Move to next chunk
>     start = end
>     end = end + chunk_size
>
> return hidden_states, past_key_values
> ```
>
> Decoding Phase
>
> ```python
> # Input: Last one token + KV cache from last iter
> # - hidden_states of that token
> # - past_key_values
>
> for layer_i in range(num_layers):
>     # Project queries just for last token
>     query, key, value = q_proj(hidden_states), k_proj(hidden_states), v_proj(hidden_states)
>
>     # First update KV cache
>     key = concat(past_key_values[layer_i].key, key)
>     value = concat(past_key_values[layer_i].value, value)
>     past_key_values[layer_i] = (key, value)
>
>     # Compute recall scores on middle section of cache (excluding global & local windows)
>     score = query @ (key[global_size:-local_size])^T
>     idx_recall = score.topk(dim=0)
>
>     # Build attention context with global + recalled + local tokens
>     key_recall = concat(
>         past_key_values[layer_i].key[:global_size],          # Global window
>         past_key_values[layer_i].key[idx_recall],            # Recalled tokens
>         past_key_values[layer_i].key[-local_size:]           # Local window
>     )
>     value_recall = concat(
>         past_key_values[layer_i].value[:global_size],
>         past_key_values[layer_i].value[idx_recall],
>         past_key_values[layer_i].value[-local_size:]
>     )
>
>     # Compute attention only for new token
>     attention_output = flash_attention(query, key_recall, value_recall)
>     hidden_states = ffn(attention_output)
>
> # Get next token prediction
> logits = lm_head(hidden_states)
> next_token = argmax(logits)
>
> return next_token, past_key_values
> ```

---

> ### Author Response · Authors · 2024-11-18
> **Response (5/5)**
>
> Regarding W6 on clarification on the "infinite" context, we appreciate your guidance on terminology. While our context length remains constrained by memory conditions, we have indeed achieved a training-free algorithm for infinite context length, representing a significant advancement over existing approaches [3,5].
>
> Regarding Q3 on differences with InfLLM, thank you for encouraging a comprehensive comparison. We have already addressed distinctions in our method and related work (subsection 2.1 line 129, subsection 2.2 line 157, section 5 line 522). The key differences between our ReAttention with InfLLM can be summarized as follows, and we will include this part in the appendix.
>   1. First, on KV cache filtering, InfLLM uses block-based representation, pre-dividing KV caches into blocks [3]. However, Existing research shows this approach can negatively impact downstream tasks by breaking context coherence [9]. ReAttention directly selects the K cache, avoiding these issues.
>   2. Second, on positional embedding, InfLLM applies identical position indices for areas beyond adjacent parts, which significantly deviates from pre-training position encoding patterns, leading to cumulative errors in the KV cache, and potentially degrading downstream task performance. We've demonstrated this in subsection 3.1's Table 1 and our previous response.
>   3. Finally, regarding efficiency tradeoff, InfLLM emphasizes offloading to reduce memory consumption [3], while our approach focuses on efficient KV cache selection via the Triton kernel [8]. Inspired by FlashAttention [7], we extract selected intermediate KV caches without storing intermediate results and achieve computation time close to the original FullAttention with lower peak memory usage.
>
> Regarding Q4 on figure readability, we will adjust the text size in figures to ensure clarity in the next version of our paper.
>
> We sincerely appreciate your detailed and constructive feedback.
>
> Best regards
>
> Authors
>
> [1] Xiao, Guangxuan, et al. "Efficient Streaming Language Models with Attention Sinks." The Twelfth International Conference on Learning Representations.
>
> [2] Zhang, Xinrong, et al. "∞ Bench: Extending long context evaluation beyond 100k tokens." Proceedings of the 62nd Annual Meeting of the Association for Computational Linguistics (Volume 1: Long Papers). 2024.
>
> [3] Xiao, Chaojun, et al. "Infllm: Training-free long-context extrapolation for llms with an efficient context memory." The Thirty-eighth Annual Conference on Neural Information Processing Systems. 2024.
>
> [4] Bai, Yushi, et al. "Longbench: A bilingual, multitask benchmark for long context understanding." arXiv preprint arXiv:2308.14508 (2023).
>
> [5] Dynamically scaled rope further increases performance of long context llama with zero fine-tuning, July 2023a. https://www.reddit.com/r/LocalLLaMA/comments/14mrgpr/dynamically_scaled_rope_further_increases/.
>
> [6] Needle in a haystack - pressure testing llms. https://github.com/gkamradt/LLMTest_NeedleInAHaystack, 2023
>
> [7] Dao, Tri. "FlashAttention-2: Faster Attention with Better Parallelism and Work Partitioning." The Twelfth International Conference on Learning Representations.
>
> [8] Tillet, Philippe, Hsiang-Tsung Kung, and David Cox. "Triton: an intermediate language and compiler for tiled neural network computations." Proceedings of the 3rd ACM SIGPLAN International Workshop on Machine Learning and Programming Languages. 2019.
>
> [9] Luo, Kun, et al. "BGE Landmark Embedding: A Chunking-Free Embedding Method For Retrieval Augmented Long-Context Large Language Models." arXiv preprint arXiv:2402.11573 (2024).

---

> ### Author Response · Authors · 2024-11-24
> **Looking forward to receiving your feedback**
>
> Dear Reviewer fRQF,
>
> We hope we have adequately addressed your issues. We would be very grateful if you could provide feedback on our rebuttal since the deadline is approaching. If you require further clarification or have any additional concerns, please do not hesitate to contact us. We are more than willing to continue communicating with you.
>
> Best wishes,
>
> The Authors

---

> ### Comment · Reviewer_fRQF · 2024-11-25
>
> Thanks to the authors for the comprehensive response, it has addressed many of my concerns (missing benchmarks and decoding throughput comparisons). Please include these results in the final version of the paper. I have raised my score to a 6.
>
> Comments:
> - From the LongBench table, it seems like the base LLaMA model achieves better performance than ReAttention. Why do you think this is the case?
> - I still don't agree with the use of the word "infinite" in the title. In practice, the method cannot support infinite sequence lengths due to memory constraints, so in what sense has "infinite context length" been achieved?

---

> > ### Author Response · Authors · 2024-11-26
> > **Response**
> >
> > Dear Reviewer,
> >
> > We sincerely appreciate your thoughtful comments and engagement with our work.
> >
> > Regarding C1, we acknowledge your observation about full attention's performance in LongBench for LLaMA. It's important to note that LongBench's 32k context length remains close to or within the model's pre-training context window. However, ReAttention's advantages become progressively more pronounced as context length increases. For instance, in InfiniteBench tests spanning 32k, 64k, and 128k contexts, our method demonstrates increasingly significant improvements, as shown in Table 1 in our paper.
> >
> > Regarding C2, while our method is constrained by memory limitations, we view these constraints as opportunities for engineering optimization. Potential strategies include memory or disk offloading to transform algorithmic infinite context into practical infinite context, without strictly prioritizing computational complexity. We are currently exploring data compression techniques to limit KV cache growth. We believe ReAttention represents a crucial step toward realizing a truly infinite context for pre-trained Transformers.
> >
> > Best wishes,
> >
> > The Authors

---

### Official Review · Reviewer_iiPf · 2024-11-04

**Soundness:** 3
**Presentation:** 3
**Contribution:** 3
**Rating:** 8
**Confidence:** 3

**Summary:**

This paper proposes ReAttention, a training-free approach to extend the context window of Large Language Models (LLMs) through a position-agnostic top-k attention mechanism. The key idea is to perform cache selection before regular self-attention, treating the extraction of critical context info as an additional attention process. The method enables models to handle theoretically infinite contexts with a finite attention scope. The authors validate their approach on multiple benchmarks and models, showing strong performance and efficiency gains through custom Triton kernel optimization.

**Strengths:**

- The problem is highly relevant and timely, as context length remains a critical bottleneck for LLMs. The paper clearly articulates three key conditions for infinite context extension (lines 040-046).
- The proposed method is simple and training-free. The two-stage attention approach (position-agnostic selection followed by regular attention) is intuitive and well-motivated.
- The empirical results are impressive. They demonstrate: (1) Comparable or better performance vs full attention across multiple benchmarks (Fig 2); (2) Successful extension of LLaMA3.2-3B context by 128x to 4M tokens (Section 3.3); (3) Strong performance on challenging tasks like LongBench, L-Eval, and InfiniteBench.
- The experimental evaluation is comprehensive, including multiple model architectures (LLaMA, Mistral, Qwen series) and various benchmarks (LongBench, L-Eval, InfiniteBench). It also includes detailed ablation studies on hyperparameters (Table 2) and thorough efficiency analysis with Triton optimization in Section 3.4.

**Weaknesses:**

- The paper has several writing issues and typos that should be addressed: e.g., Llama vs LLaMA and the wrong quotation in line 067.
- The related work section omits several relevant recent papers on efficient attention and KV-cache optimization, such as:
SnapKV [1] for efficient cache management, PyramidKV [2,3] for hierarchical cache structures, GemFilter [4] for attention-filtering, and many so on. These works tackle similar challenges and a comparison would strengthen the paper. It would be good to discuss them.

[1] Li, Yuhong, Yingbing Huang, Bowen Yang, Bharat Venkitesh, Acyr Locatelli, Hanchen Ye, Tianle Cai, Patrick Lewis, and Deming Chen. "Snapkv: Llm knows what you are looking for before generation." arXiv preprint arXiv:2404.14469 (2024).

[2] Zhang, Yichi, Bofei Gao, Tianyu Liu, Keming Lu, Wayne Xiong, Yue Dong, Baobao Chang, Junjie Hu, and Wen Xiao. "PyramidKV: Dynamic KV Cache Compression based on Pyramidal Information Funneling." arXiv preprint arXiv:2406.02069 (2024).

[3] Yang, Dongjie, XiaoDong Han, Yan Gao, Yao Hu, Shilin Zhang, and Hai Zhao. "PyramidInfer: Pyramid KV Cache Compression for High-throughput LLM Inference." arXiv preprint arXiv:2405.12532 (2024).

[4] Shi, Zhenmei, Yifei Ming, Xuan-Phi Nguyen, Yingyu Liang, and Shafiq Joty. "Discovering the gems in early layers: Accelerating long-context llms with 1000x input token reduction." arXiv preprint arXiv:2409.17422 (2024).

**Questions:**

How does ReAttention perform on more structured tasks that require precise positional understanding (e.g., code completion, mathematical reasoning)?

---

> ### Author Response · Authors · 2024-11-18
> **Response (1/2)**
>
> Dear Reviewer iiPf
>
> We appreciate your valuable feedback and recognition of our work. We have addressed your concerns as follows.
>
> Regarding W1 on writing issues and typos, we sincerely appreciate your meticulous review of our manuscript. We will carefully address and correct the spelling errors in the subsequent version of our paper.
>
> Regarding W2 on more related work, we have comprehensively expanded our discussion on KV cache optimization, especially on token eviction methods like H2O [1] and SnapKV [2]. The evolution of token eviction techniques begins with DCP [3], Scissorhands [4], and H2O [1]. These approaches demonstrate that retaining only a small subset of tokens with high attention scores could maintain near-equivalent performance while significantly reducing memory and computational overhead. Then, StreamingLLM [5] advances this concept by identifying that critical tokens always occur in the beginning and end part of the attention scope. By preserving only corresponding tokens, StreamingLLM enables stable model outputs after infinite input lengths, dramatically reducing computational and memory costs. However, this approach substantially compromised long-context performance [2,6]. Subsequent researches explore mitigation strategies, such as head-adaptive methods like FastGen [7] and DuoAttention [6]. Specifically, SnapKV [2] introduced a more nuanced approach to pruning previous KV caches by querying the above context with later input segments, substantially enhancing long-context performance for token eviction techniques. Building upon these foundations, researchers such as PyramidKV [8], PyramidInfer [9], Gemfilter [10], and LongGen [11] take layer-wise optimization, maintaining inference effectiveness while improving memory efficiency further. Besides, works like VATP [12] and InfiniPot [13] expand token pruning metrics beyond traditional QK dot product scoring. Our ReAttention draws inspiration from these token eviction methods, specifically applying the principle of selecting KV cache to context extrapolation, enabling comprehensive context perception throughout the inference process while computationally addressing each attention calculation with a finite context, thereby achieving algorithmically infinite context length.
>
> Regarding Q1 on performance on position-sensitive tasks, given the scarcity of code completion and mathematical reasoning tasks in long-context scenarios, we are actively seeking and adapting appropriate evaluations such as RepoQA [14] and Infinitebench.MathFind [15]. For the existing evaluation results, LongBench [16] includes code completion tasks like LCC [17] and RepoBench-p [18]. We have compared the performance of ReAttention and full attention across different models on LongBench's code tasks. The results are shown as follows. Our analysis reveals that ReAttention achieves comparable overall performance with full attention and outperforms StreamingLLM [5]. We are committed to further exploring and validating our method's effectiveness on position-sensitive tasks.

---

> ### Author Response · Authors · 2024-11-18
> **Response (2/2)**
>
> Results on code task in LongBench. All models are evaluated in 32k context length, with LLaMA3-8B-8K evaluated with Dynamic NTK.
>
> | | LCC | RepoBench-p | Avg on code |
> |-----|:-----:|:-----:|:-----:|
> | ***LLaMA3-8B-8K*** | 72.11 | 67.66 | 69.89 |
> | + StreamingLLM | 72.63 | 67.90 | 70.27 |
> | + ReAttention | 72.26 | 68.50 | 70.38 |
> | ***LLaMA3.1-8B-128K*** | 72.00 | 69.25 | 70.63 |
> | + StreamingLLM | 71.95 | 69.04 | 70.50 |
> | + ReAttention | 71.94 | 69.02 | 70.48 |
> | ***LLaMA3.1-70B-128K*** | 75.01 | 74.76 | 74.89 |
> | + StreamingLLM | 75.13 | 74.99 | 75.06 |
> | + ReAttention | 75.20 | 75.31 | 75.26 |
> | ***LLaMA3.2-3B-128K*** | 54.64 | 56.81 | 55.73 |
> | + StreamingLLM | 53.36 | 54.96 | 54.16 |
> | + ReAttention | 54.40 | 55.93 | 55.17 |
> | ***Mistral-v0.3-7B-32K*** | 66.21 | 64.05 | 65.13 |
> | + StreamingLLM | 66.54 | 63.16 | 64.85 |
> | + ReAttention | 66.47 | 64.16 | 65.32 |
> | ***Qwen2-7B-128K*** | 63.14 | 52.81 | 67.65 |
> | + StreamingLLM | 63.23 | 53.40 | 67.39 |
> | + ReAttention | 63.32 | 53.83 | 67.56 |
> | ***Qwen2-72B-128K*** | 67.47 | 67.82 | 71.15 |
> | + StreamingLLM | 67.49 | 67.28 | 70.91 |
> | + ReAttention | 67.52 | 67.59 | 71.39 |
> | ***Qwen2-1B-32K*** | 70.57 | 71.73 | 57.98 |
> | + StreamingLLM | 70.64 | 71.17 | 58.32 |
> | + ReAttention | 70.57 | 72.20 | 58.58 |
> | ***InternLM2.5-7B-1M*** | 53.45 | 51.78 | 52.62 |
> | + StreamingLLM | 53.85 | 52.09 | 52.97 |
> | + ReAttention | 53.44 | 51.99 | 52.72 |
> | ***Avg. on full attention*** | 66.06 | 64.07 | 65.07 |
> | ***Avg. on StreamingLLM*** | 66.09 | 63.77 | 64.93 |
> | ***Avg. on ReAttention*** | **66.12** | **64.28** | **65.20** |
>
> Thank you for your valuable feedback and suggestions.
>
> Best regards
>
> Authors
>
> [1] Zhang, Zhenyu, et al. "H2o: Heavy-hitter oracle for efficient generative inference of large language models." Advances in Neural Information Processing Systems 36 (2023): 34661-34710.
>
> [2] Li, Yuhong, et al. "Snapkv: Llm knows what you are looking for before generation." arXiv preprint arXiv:2404.14469 (2024).
>
> [3] Anagnostidis, Sotiris, et al. Dynamic context pruning for efficient and interpretable autoregressive transformers. Advances in Neural Information Processing Systems 36 (2024).
>
> [4] Liu, Zichang, et al. "Scissorhands: Exploiting the persistence of importance hypothesis for llm kv cache compression at test time." Advances in Neural Information Processing Systems 36 (2024).
>
> [5] Xiao, Guangxuan, et al. "Efficient Streaming Language Models with Attention Sinks." The Twelfth International Conference on Learning Representations.
>
> [6] Xiao, Guangxuan, et al. "DuoAttention: Efficient Long-Context LLM Inference with Retrieval and Streaming Heads." arXiv preprint arXiv:2410.10819 (2024).
>
> [7] Ge, Suyu, et al. "Model tells you what to discard: Adaptive kv cache compression for llms." arXiv preprint arXiv:2310.01801 (2023).
>
> [8] Cai, Zefan, et al. "Pyramidkv: Dynamic kv cache compression based on pyramidal information funneling." arXiv preprint arXiv:2406.02069 (2024).
>
> [9] Yang, Dongjie, et al. "PyramidInfer: Pyramid KV Cache Compression for High-throughput LLM Inference." arXiv preprint arXiv:2405.12532 (2024).
>
> [10] Shi, Zhenmei, et al. "Discovering the gems in early layers: Accelerating long-context llms with 1000x input token reduction." arXiv preprint arXiv:2409.17422 (2024).
>
> [11] Ge, Suyu, et al. "A little goes a long way: Efficient long context training and inference with partial contexts." arXiv preprint arXiv:2410.01485 (2024).
>
> [12] Guo, Zhiyu, Hidetaka Kamigaito, and Taro Watanabe. "Attention Score is not All You Need for Token Importance Indicator in KV Cache Reduction: Value Also Matters." arXiv preprint arXiv:2406.12335 (2024).
>
> [13] Kim, Minsoo, et al. "InfiniPot: Infinite Context Processing on Memory-Constrained LLMs." arXiv preprint arXiv:2410.01518 (2024).
>
> [14] Liu, Jiawei, et al. RepoQA: Evaluating Long Context Code Understanding. arXiv preprint arXiv:2406.06025 (2024).
>
> [15] Zhang, Xinrong, et al. ∞ Bench: Extending long context evaluation beyond 100k tokens. Proceedings of the 62nd Annual Meeting of the Association for Computational Linguistics (Volume 1: Long Papers). 2024.
>
> [16] Bai, Yushi, et al. Longbench: A bilingual, multitask benchmark for long context understanding. arXiv preprint arXiv:2308.14508 (2023).
>
> [17] Daya Guo, Canwen Xu, Nan Duan, Jian Yin, and Julian McAuley. 2023. Longcoder: A long-range pretrained language model for code completion. arXiv preprint arXiv:2306.14893.
>
> [18] Tianyang Liu, Canwen Xu, and Julian McAuley. 2023b. Repobench: Benchmarking repository-level code auto-completion systems. arXiv preprint arXiv:2306.03091.

---

> > ### Comment · Reviewer_iiPf · 2024-11-27
> >
> > Thank you for your response. I tend to keep my positive score.
> >
> > On the other hand, I think other reviewers' comments are valuable. I recommend that the author formally organize the new experiments in the draft. Also, I suggest the author release their code for results verification. I also suggest that the author release the Triton code, which will benefit the community.

---

> > > ### Author Response · Authors · 2024-11-27
> > > **Response**
> > >
> > > Dear Reviewer iiPf,
> > >
> > > Thank you for your recognition and comments. We will carefully organize our paper in the later stage, add new experiments, release our code timely, and contribute our Triton code to the community.
> > >
> > > Best wishes,
> > >
> > > The Authors

---

### Official Review · Reviewer_xrHG · 2024-11-04

**Soundness:** 3
**Presentation:** 2
**Contribution:** 1
**Rating:** 5
**Confidence:** 5

**Summary:**

This paper proposes ReAttention, a a training-free approach enabling LLM based on the self-attention mechanism to support an infinite context with a finite attention scope under sufficient memory resources. ReAttention splits the tokens into three parts: initial tokens, middle tokens, and local tokens. It maintains the initial and local tokens (their KV caches), select a subset of middle tokens, and re-concatenates them with positional index calibration, and then conduct attention operation.

**Strengths:**

1.	The method is intuitive. ReAttention maintains KV caches of tokens that are more frequently attended by the latest tokens, which reduces the memory overheads and remove redundant contextual segments from being attended.
2.	The evaluation is comprehensive. Authors provide evaluation for many models and benchmarks.
3.	The performance improvement is notable, compared to InfLLM and full attention.

**Weaknesses:**

1.	The two-round attention operation introduces non-trivial overheads, which may result in latency similar to full attention.
2.	The efficiency in terms of latency and throughput can be the major drawbacks of this method.
3.	The method requires sufficient memory resources where full attention is also feasible. The advantage over full attention lies in evicting irrelevant tokens, similar to token-dropping methods, which improves performance. However, efficiency in long-sequence tasks remains a concern. When full attention isn't feasible, the two-round attention calculation might be costly, as the first round still requires full-attention calculations.
4.	In long-sequence generation, the query tokens can change dynamically. How does ReAttention handle middle-cache selection in this setting? Does it re-select middle caches periodically?

**Questions:**

1.	The idea appears similar to token eviction methods like H2O and FastGen. How does ReAttention compare with these methods in terms of accuracy and efficiency?

---

> ### Author Response · Authors · 2024-11-18
> **Response (1/3)**
>
> Dear Reviewer xrHG
>
> We thank you for your valuable feedback and have addressed your concerns as follows.
>
> Regarding W1 on latency concerns, ReAttention and full attention have comparable latency, but with a critical difference, while full attention encounters Out-of-Memory (OOM) issues around 100K tokens on a single A100/800, ReAttention can continue running, as demonstrated in Figure 7. Despite additional algorithmic overhead, ReAttention proves to be more efficient practically.
>
> Regarding W2, we compared our FlashAttention2-optimized ReAttention against standard FlashAttention2 [1] implementations, looking at the throughput during decoding on a single A800 GPU. These new results complement the TTFT (Time To First Token) metrics presented in our paper.
>
> Results on decoding phase. The hyperparameters such as local size and global size used in our experiments, as well as the experimental hardware, remain consistent with the configurations we previously used in the paper. As in the paper, we compare against the official Hugging Face modeling of the Llama3 8B model with Flash Attention 2 enabled; to test decoding independently, we made minor adjustments to the code. The generation length is exactly 1024.
>
> | Context Length | Our Method (token/s) | Baseline (token/s) | batch size (ours/baseline) |
> | -------------- | ----------------------- | -----------| ---- |
> | 32k | 43.380 | 40.595 | 8/2 |
> | 64k | 24.383 | 20.399 | 5/1 |
> | 96k | 17.005 | OOM | 4/- |
> | 128k | 11.683 | OOM | 2/- |
> | 192k | 7.2813 | OOM | 1/- |
> | 256k | 5.9680 | OOM | 1/- |
>
> It's worth noting that we only utilized approximately 60% of a single A800's memory at 256k context length, without any offloading. The results show performance matching or exceeding FlashAttention, and we believe there's still untapped potential, particularly once inference engine(e.g., vLLM) optimizations are implemented.
>
> Regarding W3 on storage and computational efficiency, contrary to the reviewer's statement that "The method requires sufficient memory resources where full attention is also feasible", our ReAttention uniquely enables operation beyond 100K tokens on a single A100, as shown in Figure 7. This is achieved through chunk-based reading to reduce peak memory utilization and a custom kernel for TopK Attention that minimizes intermediate computation memory read/write. These strategies are detailed in section 3.4.

---

> ### Author Response · Authors · 2024-11-18
> **Response (2/3)**
>
> Regarding W4 on generation process details, we appreciate the reviewer's concern. Indeed, our approach implements dynamic token filtering at each generation step, as detailed in Section 2.1. We've drafted detailed pseudocode showing how ReAttention works during both prefilling and decoding. We plan to include this in the appendix to make the algorithm more accessible to readers. For now, we've prepared a markdown version (due to website rendering problems for Latex) for your review – would love to get your thoughts on it.
>
> Prefilling Phase
>
> ```python
> # Input: seq_len tokens
> # Configuration:
> #   - global_size tokens at start of cache
> #   - local_size tokens at end of cache
> #   - chunk_size for processing
>
> start = 0
> end = global_size + local_size
> past_key_values = None
>
> while start < seq_len:
>     # Process next chunk
>     input_chunk = input_ids[:, start:min(end, seq_len)]
>     hidden_states = embedding(input_chunk)
>
>     # Forward through each transformer layer
>     for layer_i in range(num_layers):
>         # Project to Q,K,V
>         query = q_proj(hidden_states)
>         key = k_proj(hidden_states)
>         value = v_proj(hidden_states)
>
>         if layer_i == 0:
>             # Apply position embeddings
>             position_ids = get_position_ids(start, end)
>             cos, sin = get_rope_embeddings(position_ids)
>             query, key = apply_rope(query, key, cos, sin)
>
>         # Update KV cache for this layer
>         if past_key_values:
>             key = concat(past_key_values[layer_i].key, key)
>             value = concat(past_key_values[layer_i].value, value)
>         past_key_values[layer_i] = (key, value)
>
>         score = query @ (key[global_size:-local_size])^T
>               idx_recall = score.topk(dim=0)
>
>         key_recall = concat(past_key_values[layer_i].key[global_size:],
>                            past_key_values[layer_i].key[idx_recall],
>                            past_key_values[layer_i].key[:-local_size])
>         value_recall = concat(past_key_values[layer_i].value[global_size:],
>                            past_key_values[layer_i].value[idx_recall],
>                            past_key_values[layer_i].value[:-local_size])
>
>         # Compute attention
>         attention_output = flash_attention(query, key_recall, value_recall)
>         hidden_states = ffn(attention_output)
>
>     # Move to next chunk
>     start = end
>     end = end + chunk_size
>
> return hidden_states, past_key_values
> ```
>
> Decoding Phase
>
> ```python
> # Input: Last one token + KV cache from last iter
> # - hidden_states of that token
> # - past_key_values
>
> for layer_i in range(num_layers):
>     # Project queries just for last token
>     query, key, value = q_proj(hidden_states), k_proj(hidden_states), v_proj(hidden_states)
>
>     # First update KV cache
>     key = concat(past_key_values[layer_i].key, key)
>     value = concat(past_key_values[layer_i].value, value)
>     past_key_values[layer_i] = (key, value)
>
>     # Compute recall scores on middle section of cache (excluding global & local windows)
>     score = query @ (key[global_size:-local_size])^T
>     idx_recall = score.topk(dim=0)
>
>     # Build attention context with global + recalled + local tokens
>     key_recall = concat(
>         past_key_values[layer_i].key[:global_size],          # Global window
>         past_key_values[layer_i].key[idx_recall],            # Recalled tokens
>         past_key_values[layer_i].key[-local_size:]           # Local window
>     )
>     value_recall = concat(
>         past_key_values[layer_i].value[:global_size],
>         past_key_values[layer_i].value[idx_recall],
>         past_key_values[layer_i].value[-local_size:]
>     )
>
>     # Compute attention only for new token
>     attention_output = flash_attention(query, key_recall, value_recall)
>     hidden_states = ffn(attention_output)
>
> # Get next token prediction
> logits = lm_head(hidden_states)
> next_token = argmax(logits)
>
> return next_token, past_key_values
> ```

---

> ### Author Response · Authors · 2024-11-18
> **Response (3/3)**
>
> Regarding Q1 on token eviction methods,  we emphasize that we are not a KV cache optimization method. We do not perform KV cache eviction [2], but instead select critical tokens before self-attention computation, thereby achieving algorithmically infinite context. Considering the reviewer's interest in downstream task performance, we conduct a comparative analysis with H2O [2] on LongBench [3].
>
>  Results on LongBench. All experiments are conducted in 32k context length, while LLaMA3-8B-8K is evaluated with Dynamic NTK.
>
> | | En | Zh | Code | Avg. |
> |-----|:-----:|:-----:|:-----:|:-----:|
> | ***LLaMA3-8B-8K*** | 34.27 | 19.95 | 69.89 | 35.48 |
> | + StreamingLLM | 29.48 | 13.22 | 70.27 | 32.49 |
> | + InfLLM | 29.53 | 17.60 | 70.03 | 32.51 |
> | + H2O | 29.68 | 16.33 | 69.40 | 33.08 |
> | + ReAttention | 33.42 | 18.74 | 70.38 | 35.03 |
> | ***LLaMA3.1-8B-128K*** | 36.38 | 28.64 | 70.63 | 38.79 |
> | + StreamingLLM | 33.41 | 24.24 | 70.50 | 37.06 |
> | + InfLLM | 34.26 | 27.22 | 68.37 | 37.14 |
> | + H2O | 35.66 | 28.26 | 70.69 | 38.32 |
> | + ReAttention | 36.14 | 28.56 | 70.48 | 38.63 |
>
> Best regards
>
> Authors
>
> [1] Dao, Tri. "FlashAttention-2: Faster Attention with Better Parallelism and Work Partitioning." The Twelfth International Conference on Learning Representations.
>
> [2] Zhang, Zhenyu, et al. "H2O: Heavy-hitter oracle for efficient generative inference of large language models." Advances in Neural Information Processing Systems 36 (2023): 34661-34710.
>
> [3] Bai, Yushi, et al. Longbench: A bilingual, multitask benchmark for long context understanding. arXiv preprint arXiv:2308.14508 (2023).

---

> ### Author Response · Authors · 2024-11-24
> **Looking forward to receiving your feedback**
>
> Dear Reviewer xrHG,
>
> We hope we have adequately addressed your issues. We would be very grateful if you could provide feedback on our rebuttal since the deadline is approaching. If you require further clarification or have any additional concerns, please do not hesitate to contact us. We are more than willing to continue communicating with you.
>
> Best wishes,
>
> The Authors

---

> ### Comment · Reviewer_xrHG · 2024-11-26
>
> I appreciate the authors' detailed rebuttal and the additional insights provided regarding ReAttention's capabilities and limitations. While I acknowledge that ReAttention addresses the memory-bound problem effectively, allowing previously intractable sequence lengths to become tractable, I believe its approach shares conceptual similarities with token-dropping methods like H2O and FastGen. Specifically, selecting and attending to critical tokens is functionally equivalent to maintaining their KV cache for use during the decoding phase. This strategy effectively reduces memory overhead by discarding less important tokens, which aligns with the principles of token eviction techniques.
>
> However, a significant drawback of ReAttention remains its latency. The two-round attention calculation introduces notable computational overhead. The additional throughput results in rebuttal does not address my concerns but somehow validate my concerns due to the trade-off between latency and throughput when increasing batch size. As shown by the throughput results, while ReAttention enables up to a 5x larger batch size compared to full attention, the throughput between the two methods is nearly equivalent. This indicates that the latency (of generating one token) of ReAttention is approximately 5x higher than that of full attention.  Can authors fix the sequence length (e.g, 32k) and show the comparison to full attention about figure of latency (or throughput) v.s. batch size? The figure of latency v.s. batch size can show the efficiency of ReAttention and can be done given different fixed length (32k and 64k).  Given that full attention already imposes substantial latency and computational complexity, ReAttention’s additional latency represents a considerable bottleneck that undermines its overall efficiency. Additionally, I encourage the authors to provide a comprehensive accuracy comparison between full attention to establish a more holistic understanding of the trade-offs between memory efficiency, latency, and predictive performance.
>
> To make ReAttention more practical and widely applicable, I believe its computational complexity and latency need to be further optimized. These improvements would significantly enhance the applicability of ReAttention for long-sequence tasks.

---

> > ### Author Response · Authors · 2024-11-27
> > **Response (2/3)**
> >
> > Regarding a comprehensive accuracy comparison, we summarize the downstream evaluation we have done between ReAttention and full attention. Firstly, in 32k context length, we report the results of LongBench and LEval, which demonstrate that ReAttention maintains the accuracy of full attention while extending its context length.
> >
> > Results of LongBench & LEval. LLaMA3-8B-8K are evaluated with DynamicNTK, scaling_factor=4.
> >
> > | | LongBench | | LEval | |
> > |----|:----:|:----:|:----:|:----:|
> > | | **Full Attention** | **ReAttention** | **Full Attention** | **ReAttention** |
> > | ***LLaMA3-8B-8K*** | 35.48 | 35.03 | 17.23 | 16.60 |
> > | ***LLaMA3.1-8B-128K*** | 38.79 | 38.63 | 15.44 | 15.42 |
> > | ***LLaMA3.1-70B-128K*** | 46.02 | 46.15 | 20.36 | 20.49 |
> > | ***LLaMA3.2-3B-128K*** | 40.76 | 39.43 | 32.33 | 31.11 |
> > | ***Mistral-v0.3-7B-32K*** | 35.61 | 35.48 | 16.88 | 16.71 |
> > | ***InternLM2.5-7B-1M*** | 47.27 | 47.35 | 37.64 | 37.40 |
> > | ***Qwen2-7B-128K*** | 41.34 | 41.01 | 28.08 | 27.96 |
> > | ***Qwen2-72B-128K*** | 54.71 | 54.53 | 39.68 | 39.33 |
> > | ***Qwen2-1B-32K*** | 33.21 | 33.51 | 24.16 | 24.40 |
> > | ***Avg.*** | 41.46 | 41.23 | 25.75 | 25.47 |
> >
> > Moreover, ReAttention demonstrates a clearer advantage over full attention as context length increases, as shown in InfiniteBench results in 32k, 64k, and 128k contexts.
> >
> > Results of InfiniteBench. LLaMA3-8B-Instruct-8K, Qwen2-7B-Instruct-32K, and Qwen2-1B-Instruct-32K are evaluated with DynamicNTK, scaling_factor=4.
> >
> > |   |   | 32k |   |   | 64k |   |   | 128k |   | Avg. |
> > |-----|:-----:|:-----:|:-----:|:-----:|:-----:|:-----:|:-----:|:-----:|:-----:|:-----:|
> > |   | **MC** | **QA** | **Sum** | **MC** | **QA** | **Sum** | **MC** | **QA** | **Sum** |   |
> > | ***LLaMA3-8B-Instruct-8K*** | 50.66 | 7.44 | 21.76 | — | — | — | — | — | — | — |
> > | + ReAttention | 46.29 | 5.33 | 21.76 | 44.54 | 5.25 | 20.15 | 49.78 | 5.98 | 20.36 | **24.38** |
> > | ***LLaMA3.1-8B-Instruct-128K*** | 37.12 | 18.85 | 1.60 | 31.88 | 23.32 | 1.39 | 21.83 | 25.43 | 1.40 | 18.09 |
> > | + ReAttention | 36.68 | 12.71 | 19.68 | 35.37 | 12.14 | 18.84 | 40.61 | 12.63 | 18.14 | **22.98** |
> > | ***LLaMA3.2-3B-Instruct-128K*** | 14.41 | 15.65 | 1.38 | 11.79 | 18.16 | 1.27 | 13.10 | 17.63 | 1.30 | 10.52
> > | + ReAttention | 19.21 | 11.80 | 16.75 | 17.47 | 11.82 | 15.06 | 20.09 | 11.91 | 14.13 | **15.36** |
> > | ***Qwen2-7B-Instruct-32K*** | 35.37 | 2.81 | 1.69 | 31.88 | 6.18 | 1.67 | 27.95 | 3.75 | 1.77 | 12.56 |
> > | + ReAttention | 35.37 | 2.17 | 16.30 | 35.81 | 1.70 | 16.63 | 39.74 | 1.50 | 15.78 | **18.33** |
> > | ***Qwen2-1B-Instruct-32K*** | 35.37 | 3.65 | 1.79 | 28.38 | 4.97 | 1.62 | 26.20 | 4.87 | 1.65 | 12.06 |
> > | + ReAttention | 41.92 | 3.04 | 13.24 | 38.43 | 2.55 | 11.93 | 40.61 | 3.54 | 12.43 | **18.63** |

---

> > ### Author Response · Authors · 2024-11-27
> > **Response (3/3)**
> >
> > Furthermore, our Needle-in-a-Haystack evaluation in LLaMA3-8B-8K also reveals that while full attention with dynamic NTK struggles beyond 50K tokens and fails at 100K due to OOM, ReAttention maintains perfect retrieval accuracy up to 400K tokens across most depth levels. Both results are evaluated in one A800 GPU, which proves that ReAttention could extend the context length without sacrificing accuracy and memory usage.
> >
> > Results of NIAH
> >
> > LLaMA3-8B-8K with DynamicNTK, scaling_factor=4
> >
> > | depth \ length | 8k | 16k | 32k | 50k | 100k |
> > |-----|:-----:|:-----:|:-----:|:-----:|:-----:|
> > | 0% | 100.00 | 100.00 | 100.00 | 3.19 | OOM |
> > | 10% | 100.00 | 100.00 | 100.00 | 3.23 | OOM |
> > | 20% | 100.00 | 100.00 | 100.00 | 12.95 | OOM |
> > | 30% | 100.00 | 100.00 | 100.00 | 3.20 | OOM |
> > | 40% | 100.00 | 100.00 | 100.00 | 3.43 | OOM |
> > | 50% | 100.00 | 100.00 | 100.00 | 3.28 | OOM |
> > | 60% | 100.00 | 100.00 | 90.35 | 3.42 | OOM |
> > | 70% | 100.00 | 100.00 | 100.00 | 3.46 | OOM |
> > | 80% | 100.00 | 100.00 | 100.00 | 3.33 | OOM |
> > | 90% | 100.00 | 100.00 | 100.00 | 3.52 | OOM |
> > | 100% | 100.00 | 100.00 | 100.00 | 22.51 | OOM |
> >
> > LLaMA3-8B-8K with ReAttention, local_size=4096, global_size=32, seg_size=32, num_seg=127=k', chunk_size=512
> >
> > | depth \ length | 8k | 16k | 32k | 50k | 100k | 200k | 300k | 400k |
> > |-----|:-----:|:-----:|:-----:|:-----:|:-----:|:-----:|:-----:|:-----:|
> > | 0% | 100.00 | 100.00 | 100.00 | 100.00 | 100.00 | 100.00 | 100.00 | 100.00 |
> > | 10% | 100.00 | 100.00 | 100.00 | 100.00 | 100.00 | 90.42 | 100.00 | 100.00 |
> > | 20% | 100.00 | 100.00 | 100.00 | 100.00 | 100.00 | 100.00 | 100.00 | 100.00 |
> > | 30% | 100.00 | 100.00 | 100.00 | 100.00 | 100.00 | 100.00 | 100.00 | 100.00 |
> > | 40% | 100.00 | 100.00 | 100.00 | 100.00 | 100.00 | 90.40 | 100.00 | 100.00 |
> > | 50% | 100.00 | 100.00 | 100.00 | 100.00 | 100.00 | 100.00 | 100.00 | 100.00 |
> > | 60% | 100.00 | 100.00 | 100.00 | 100.00 | 100.00 | 100.00 | 100.00 | 100.00 |
> > | 70% | 100.00 | 100.00 | 100.00 | 100.00 | 100.00 | 100.00 | 100.00 | 100.00 |
> > | 80% | 100.00 | 100.00 | 100.00 | 100.00 | 100.00 | 90.29 | 90.42 | 100.00 |
> > | 90% | 100.00 | 100.00 | 100.00 | 100.00 | 100.00 | 90.42 | 100.00 | 100.00 |
> > | 100% | 100.00 | 100.00 | 100.00 | 100.00 | 100.00 | 100.00 | 100.00 | 100.00 |
> >
> > More results of our NIAH evaluation can be found in Fig 3 and Fig 4 in our paper. We hope these results demonstrate that ReAttention preserves full attention's accuracy while expanding its context length. We hope these results in LongBench, LEval, InfiniteBench, and NIAH could address your concern about comprehensive accuracy comparisons.
> >
> > We remain open to further constructive dialogue and appreciate the opportunity to refine our work.
> >
> > Best wishes,
> >
> > The Authors

---

> ### Author Response · Authors · 2024-11-27
> **Response (1/3)**
>
> Dear Reviewer xrHG,
>
> We sincerely thank you for your continued engagement and thoughtful feedback.
>
> Regarding the comparison to token dropping or eviction methods, there are differences in both approach and capability that need clarification. While token-dropping methods like H2O and FastGen permanently discard tokens, ReAttention preserves the complete context in the top-k attention. Token-dropping methods make irreversible decisions about which information to discard, creating an inherent shortage of losing potentially valuable context that cannot be recovered later. ReAttention, on the other hand, implements a dynamic selection mechanism during computation. All tokens remain available in the KV cache, and our approach determines which tokens to attend to during each specific computation step. This means the model retains access to the full context and can flexibly select different subsets of tokens in each attention head, layer, and inference step. **ReAttention could see the token unseen in the previous inference step, but H2O fails to see the token dropped before.** Therefore, ReAttention extends the context length to up to 1M tokens with perfect retrieval accuracy on NIAH tasks, where token-dropping methods like H2O and FastGen fail to do so.
>
> Regarding the comparison between full attention and ReAttention in efficiency, the throughput and TTFT (Time To First Token) are two different measuring dimensions. TTFT measures performance under identical conditions (same batch size and context length). Our TTFT results demonstrate that ReAttention achieves latency on par with full attention equipped with FlashAttention2 under these controlled conditions.
>
> Using a single A800 GPU, we test our ReAttention approach and the standard implementation of Hugging Face's **Llama 3 8B modeling with FlashAttention2. We keep all the settings, e.g. local size and global size, the same as in our paper. The batch size is exactly 1.
>
> | Context Length | Our Method (ms) | Baseline (ms) |
> |---------|----------------|-------------------|
> | 32768 | 3984.66 | 3592.32 |
> | 65536 | 10696.22 | 9823.82 |
> | 98304 | 20139.42 | 18942.57 |
> | 131072 | 32328.20 | 31953.82 |
> | 163840 | 47156.63 | 46322.93 |
> | 196608 | 64832.10 | 64475.56 |
> | 229376 | 85098.45 | 0.00(OOM) |
> | 262144 | 108240.09 | 0.00(OOM) |
>
> Throughput, on the other hand, evaluates maximum processing capacity when fully utilizing available GPU memory. ReAttention demonstrates higher efficiency by supporting larger batch sizes while maintaining comparable or better token processing rates. Besides, beyond context length 64K, full attention equipped with FlashAttention2 encounters memory limitations, while ReAttention continues working.
>
> Using a single A800 GPU, we test our ReAttention approach and full attention with FlashAttention2 with LLaMA3-8B. We keep all the settings, e.g. local size and global size, the same as in our paper. The generation length is exactly 1024.
>
> | Context Length | Our Method (token/s) | Baseline (token/s) | Batch Size (ours/baseline) |
> | -------------- | ----------------------- | -----------| ---- |
> | 32k | 43.380 | 40.595 | 8/2 |
> | 64k | 24.383 | 20.399 | 5/1 |
> | 96k | 17.005 | OOM | 4/- |
> | 128k | 11.683 | OOM | 2/- |
> | 192k | 7.2813 | OOM | 1/- |
> | 256k | 5.9680 | OOM | 1/- |
>
> This performance pattern indicates not a *trade-off* where we sacrifice one capability for another. Rather, it represents extending context length while maintaining competitive throughput and enabling larger batch sizes.

---

### Meta-Review · Area_Chair_kqsT · 2024-12-20

**Metareview:**

The paper proposes ReAttention, a novel training-free method that enables large language models (LLMs) to handle effectively infinite context lengths with a finite attention scope. The key innovation lies in performing a position-agnostic top-k attention before the regular self-attention mechanism, allowing ReAttention to dynamically manage and retrieve essential context tokens. The authors validate the approach across benchmarks, demonstrating significant performance gains in long-context scenarios. ReAttention stands out for extending context lengths of mainstream LLMs like LLaMA and Mistral to unprecedented lengths (up to 4M tokens) without requiring additional training. Additionally, the Triton-based implementation highlights the engineering contributions, which optimize computational efficiency.

Reviewers generally commended the ability of ReAttention to extrapolate context length significantly beyond pre-trained limits while maintaining competitive or superior performance. The engineering contributions, including a highly efficient Triton kernel, further solidify its applicability for real-world scenarios. The method’s flexibility in maintaining context coherence without positional embeddings is both conceptually and empirically validated. ReAttention’s capacity to extend LLMs' context length while preserving retrieval accuracy in challenging tasks is a compelling advancement in efficient and scalable LLM inference.

Overall, the reviewers converged on the paper’s merit as an impactful contribution to the field, with key innovations and broad applicability. Thus I recommend acceptance. Meanwhile, I do concur with Reviewer fRQF that the word "infinite" in the title is proper because the authors **did not demonstrate it**. The authors are advised to pick a more solid title in the final revision.

**Additional Comments On Reviewer Discussion:**

During the discussion period, the reviewers raised several important concerns, including the need for additional baseline comparisons, clarity in distinguishing ReAttention from related methods like StreamingLLM and H2O, and more detailed explanations of the method’s efficiency and effectiveness. The authors actively engaged with these points, providing new experiments comparing ReAttention with methods such as SnapKV and InfLLM on benchmarks like LongBench and InfiniteBench. These results demonstrated that ReAttention consistently matches or outperforms these baselines, particularly in long-context and retrieval-heavy tasks.

The reviewers were also concerned about the computational overhead introduced by the two-round attention mechanism. The authors provided detailed throughput analyses and Time-To-First-Token metrics, showing that ReAttention achieves competitive efficiency, especially in long-context scenarios where other methods face memory limitations. The discussion clarified that ReAttention’s advantages become pronounced as context length increases, addressing doubts about its scalability.

Although some reviewers, like bDJC, remained unconvinced about ReAttention being fundamentally distinct from StreamingLLM or H2O, the authors emphasized that ReAttention does not perform token eviction but instead dynamically selects tokens for attention without compromising context coherence, which I find makes sense to some extent.

Ultimately, most reviewers appreciated the authors’ responsiveness and efforts to address concerns, leading to score increases. The thorough empirical evaluation, coupled with the practical contributions, outweighs the minor weaknesses, supporting the final recommendation to accept the paper as a positive contribution to the field.

---

### Decision · Program_Chairs · 2025-01-22

Accept (Poster)